# Proximity labeling of DAF-16 FOXO highlights aging regulatory proteins

Murat Artan[1,2] ✉, Hanna Schoen[1] & Mario de Bono [1] ✉

Insulin/insulin-like growth factor signaling inhibits FOXO transcription factors to control development, homeostasis, and aging. Here, we use proximity labeling to identify proteins interacting with the *C. elegans* FOXO DAF-16. We show that in well-fed, unstressed animals harboring active insulin signaling, DAF-16 forms a complex with the PAR-1/MARK serine/threonine kinase, a key regulator of cell polarity. PAR-1 inhibits DAF-16 accumulation and promotes DAF-16 phosphorylation at S249, at a conserved motif that PAR-1/human MARK2 phosphorylates in vitro. DAF-2 insulin-like receptor signaling stimulates DAF-16 S249 phosphorylation, suggesting DAF-2 activates PAR-1. DAF-2 also promotes PAR-1 expression by inhibiting DAF-16. PAR-1 knockdown, or DAF-16 S249A, prolong lifespan, whereas phosphomimetic DAF-16 S249D suppresses the longevity of *daf-2* mutants. At low insulin signaling, DAF-16 proximity labeling highlights transcription factors, chromatin regulators, and DNA repair proteins. One interactor, the zinc finger/homeobox protein ZFH-2/ ZFHX3, forms a complex with DAF-16 and prolongs lifespan. Our work provides entry points for hypothesis-driven studies of FOXO function and longevity.

Animals have evolved mechanisms to re-program their physiology in the face of stresses and privations, and thus increase their longevity. FOXO/DAF-16 transcription factors (TFs) are prominent mediators of such adaptation, and play this role across metazoa[1]. FOXO activity is negatively regulated by the insulin/insulin-like growth factor signaling (IIS) pathway, and is one of the main outputs of IIS[2]. In *C. elegans (Ce)*, ample food and favorable growth conditions promote high insulin-like signaling, activating the DAF-2 insulin/IGF receptor. This triggers a kinase cascade in which activation of the phosphatidylinositol-3-kinase (PI3K) AGE-1[3,4] stimulates 3-phosphoinositide-dependent kinase 1 (PDK-1)[5], likely by increasing the $PI(3, 4, 5)P_3/PI(4, 5)P_2$ ratio[6]. PDK-1 in turn stimulates AKT-1 and AKT-2, kinases belonging to the Akt/protein kinase B (PKB) family, which phosphorylate and inhibit DAF-16/FOXO by causing its cytolasmic sequestration by 14-3-3 proteins[5,7].

Stress or starvation downregulates IIS, allowing dephosphorylated DAF-16/FOXO to enter the nucleus and alter expression of target genes that include chaperones, antioxidants, and antimicrobials[8,9]. This reprogramming increases the animal's stress resistance and prolongs its lifespan. Given the importance of FOXO functions, elucidating the different ways these TFs are regulated, and how they collaborate with co-factors to exert their many functions, have become key goals[10-14].

Several studies have used targeted co-immunoprecipitation (co-IP), or co-IP combined with proteomic analysis, to identify DAF-16 binding partners[10,13-26]. We recently optimized TurboID-mediated proximity labeling in *C. elegans*[27,28] and speculated this approach would offer a useful alternative to identify DAF-16 regulators and effectors. TurboID biotin ligase fused to DAF-16 should biotinylate proteins within a ~10 nm radius[29]. Biotin's femtomolar affinity for streptavidin permits affinity purification of these biotinylated proteins under harsh, denaturing conditions, contrasting with the gentle extraction and washing required to retain complex integrity during co-IP. TurboID's high activity allows detection of transient or weak interactors[30] and highlights proximity at any point in the life of the

[1]Institute of Science and Technology Austria (ISTA), Am Campus 1, Klosterneuburg, Austria. [2]Present address: University of Cologne, Faculty of Mathematics and Natural Sciences, Cologne Excellence Cluster for Aging and Aging-Associated Diseases (CECAD), Cologne, Germany. ✉e-mail: martan@uni-koeln.de; mdebono@ista.ac.at

tagged protein. Unlike co-IP, proximity labeling does not necessarily mean two proteins are in a complex, but in vivo proximity is often biologically meaningful.

Here, we used CRISPR to functionally tag endogenous DAF-16 with TurboID. We then performed in vivo proximity labeling of DAF-16/FOXO under normal and low IIS conditions. This allowed us to identify potential interactors of both cytosolic and nuclear DAF-16. We selected two interactors for detailed study, and these have provided insights into the control of FOXO function and longevity. Notably, we find that the cell polarity kinase PAR-1, the ortholog of human MARK kinases, participates in IIS signaling. PAR-1 forms a complex with DAF-16 in vivo and promotes its phosphorylation at serine 249 in a conserved sequence motif. Recombinant PAR-1 and human MARK2 can phosphorylate synthetic peptides corresponding to this motif in DAF-16 and human FOXO3a. PAR-1 knockdown triggers DAF-16 nuclear entry and extends lifespan, as does mutating DAF-16 S249 to alanine. Conversely, phosphomimetic DAF-16 S249D suppresses phenotypes exhibited by loss-of-function mutants of the *daf-2* insulin receptor, including constitutive dauer formation, reduced brood size, and extended lifespan. At low IIS, we find that the zinc finger and homeobox protein ZFH-2, the worm ortholog of the ZFHX3 tumor suppressor, forms a complex with DAF-16 in the nucleus and, like DAF-16, extends lifespan. Other potential DAF-16 interactors we identify merit future study.

## Results

### Using TurboID to identify potential DAF-16/FOXO interactors at different insulin/IGF receptor signaling levels

The DAF-16/FOXO transcription factor shuttles from the nucleus to the cytoplasm in response to signaling from the DAF-2 insulin/IGF receptor[31,32]. In well-fed, unstressed animals, high DAF-2 signaling keeps DAF-16 cytoplasmic. Stress suppresses DAF-2 signaling, causing DAF-16 to accumulate in the nucleus. We used TurboID-mediated proximity labeling[29] to characterize proteins that interact with DAF-16 in well-fed, unstressed, wild-type animals (cytosolic DAF-16) and in *daf-2* mutants (nuclear DAF-16).

We edited the endogenous *daf-16* gene by inserting sequences encoding TurboID::mNeongreen::3xFLAG (TbID::mNG) in frame just upstream of the stop codon (Fig. 1a and Supplementary Fig. 1a). Although *daf-16* expresses multiple isoforms, all should be tagged since they share their C-terminus. We used fluorescence microscopy to visualize DAF-16::TbID::mNG subcellular localization in wild-type and *daf-2(e1370)* hypomorphic mutants. As expected, DAF-16::TbID::mNG was predominantly cytoplasmic in a wild-type background (which we will refer to as cytoDAF-16), and nuclear in a *daf-2(e1370)* mutant background (which we refer to as nucDAF-16) (Fig. 1b). We used these strains to perform proximity labeling (see "Methods"[28]). As a control, we processed extracts from wild-type animals that did not express TurboID. This allowed us to identify and discard proteins that nonspecifically bound the affinity column under the extraction conditions we used. Applying a cut-off value of $\text{Log}_2$ 2, we identified 948 proteins significantly enriched, and 76 proteins significantly depleted ($p < 0.05$) when we compared mass spectrometry data obtained from extracts made from *daf-16::TbID::mNG* animals to data from wild-type (no TbID) extracts (Fig. 1c and Supplementary Data 1). Most proteins whose signal increased compared to the no TbID control were found in both the cytoDAF-16 and nucDAF-16 experiments (Supplementary Fig. 1b, c and Supplementary Data 1).

To focus our further studies, we next highlighted proteins whose interaction with DAF-16::TbID::mNG changed according to *daf-2* signaling status. We identified 240 potential interactors significantly enriched in nucDAF-16 vs cytoDAF-16, and 21 proteins significantly depleted for the same comparison (Fig. 1d–f and Supplementary Data 1). Gene ontology (GO) analysis categorized most proteins proximal to DAF-16 when *daf-2* signaling was low as nuclear (Fig. 1g), as

expected. These proteins included the conserved transcription factors DPFF-1/DPF2 and ZFH-2/ZFHX3, the RNA Pol II phosphatase FCP-1/CTDP1, the DNA binding protein LIN-9/LIN9, the chromatin remodeling complex subunit PBRM-1/PBRM1, the histone lysine methyltransferases SET-26/MLL5 and ZFP-1/MLLT10, and the SWI/SNF complex subunits SWSN-1/SMARCC1-2 and SWSN-4/ SMARCA2/4, with the last two previously reported to physically interact with DAF-16[10] (Fig. 1d, e and Supplementary Data 1). Grainyhead 1/GRH-1, another potential interactor, has been linked to insulin signaling and lifespan, but was not previously shown to interact with DAF-16[33]. We also identified the DNA damage recognition and repair proteins XPC-1/XPC (Xeroderma pigmentosum group C), RFC-4 (replication factor C subunit 4), and SEL-13 (ortholog of human ZNF830). These nuclear proteins may functionally interact with DAF-16 on chromatin to regulate transcription or DNA repair.

A distinct set of proteins selectively interacted with cytoplasmic DAF-16::TbID::mNG, in animals with wild-type DAF-2 signaling. These proteins included the serine/threonine kinase PAR-1 (the ortholog of human MARK kinases), which is a key regulator of cell polarity and microtubule dynamics, the MTCL1 (microtubule crosslinking factor 1) ortholog MTCL-1, the kinesins UNC-104/KIF1A and OSM-3/KIF17, the TOM1 (target of *myb1* membrane trafficking protein) ortholog TMYB-1, the vesicular transport adapter protein APT-9 (which is the *Ce* ortholog of human GGA1-3), and the presynaptic protein SYD-1 (the ortholog of human SYDE, synapse defective Rho GTPase homolog) (Fig. 1d, f and Supplementary Data 1). These proteins are implicated in vesicular traffic and microtubule function. They hint at mechanisms that move DAF-16 around the cell when insulin signaling is high, although this needs to be investigated.

By searching the literature, we identified 17 proteins that are not listed in Fig. 1e, f but are known to co-IP with DAF-16. We could detect and quantify 12 of these proteins in our proximity labeling data. Six, SIR-2.1, HCF-1, SGK-1, AKT-1/2, and PRMT-1, were significantly enriched proximal to DAF-16 compared to control (Supplementary Data 2). Two of these, SIR-2.1 and HCF-1, interacted significantly more with DAF-16 when *daf-2* was defective (Supplementary Data 2), but below the threshold for inclusion in Fig. 1e. These data help benchmark the co-IP and proximity labeling approaches.

Altered labeling of highlighted proteins by DAF-16::TbID::mNG could reflect IIS-dependent changes in their expression. To test this, we used quantitative RT-PCR (qRT-PCR) to compare mRNA expression in wild-type, *daf-2*, and *daf-2; daf-16* mutants (Supplementary Fig. 1d). We focused on 10 genes encoding potential DAF-16 interactors with human homologs. As controls we included two genes known to be regulated by DAF-2 and DAF-16, *mtl-1* (metallothionein 1) and *sod-3* (superoxide dismutase 3) (Supplementary Fig. 1e). For 5 genes, we did not observe any changes in mRNA levels (Supplementary Fig. 1d). Levels of *xpc-1* mRNA doubled in *daf-2* mutants, whereas *par-1* mRNA levels halved, and both these changes required functional DAF-16 (Supplementary Fig. 1d). We also observed a small decrease in *zfh-2* mRNA levels in *daf-2* mutants but this was only partly *daf-16*-dependent (Supplementary Fig. 1d). The modest changes in the mRNA levels of the genes tested, or lack thereof, suggests altered expression does not account for the *daf-2*-dependent changes we observed in our DAF-16 proximity labeling data.

### Characterizing putative DAF-16 interactors

To test whether proteins we identified as proximal to DAF-16 impacted DAF-16 function, we first performed an RNAi secondary screen. Metallothioneins protect against heavy metal toxicity and oxidative stress, and previous work has shown that DAF-16 promotes expression of the metallothionein *mtl-1*[8,34]. To create a reporter that recapitulates the physiological expression of *mtl-1*, we knocked in sequences encoding mNG::3xFLAG just upstream of the *mtl-1* stop codon (Supplementary Fig. 2a, b). Using available clones from the Ahringer RNAi

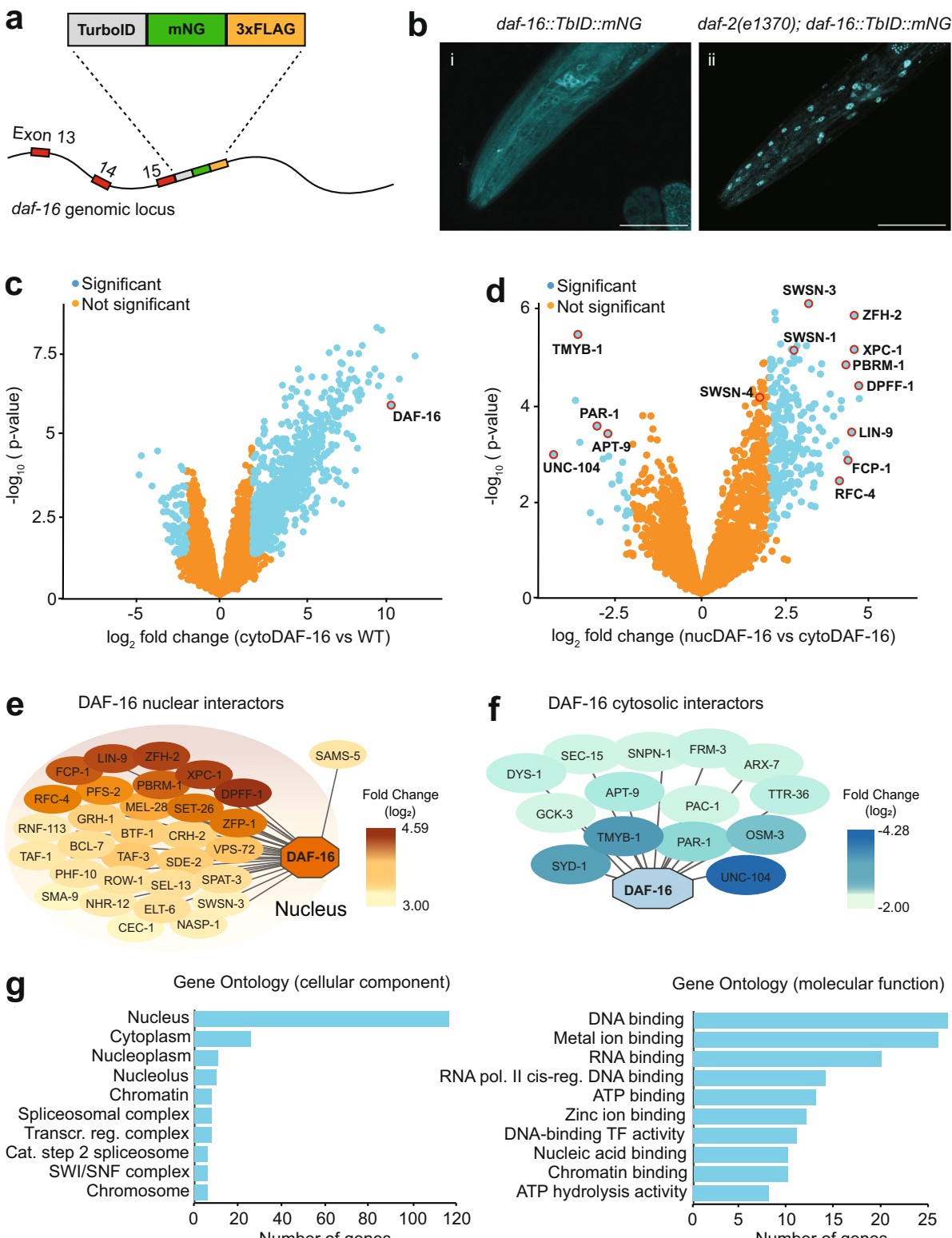

library[35], and focusing on genes conserved beyond Nematoda, we knocked down 17 potential DAF-16 interactors in WT or *daf-2(e1370)* strains that also harbored the edited *mtl-1::mNG::3xFLAG* gene. As controls, we used an empty vector and *daf-16* RNAi. To quantify changes in MTL-1::mNG::3xFLAG expression, we used Western blots. RNAi-mediated knockdown of *pfs-2, par-1, zfh-2, mtcl-1,* and *lin-9* increased MTL-1::mNG::3xFLAG expression in both wild-type (Fig. 2a,

b) and *daf-2(e1370)* mutant backgrounds (Supplementary Fig. 2c, d). By contrast, knocking down *dpff-1* decreased MTL-1::mNG::3xFLAG expression in both tested backgrounds (Fig. 2a, b and Supplementary Fig. 2c, d). For other knockdowns, we only observed significantly altered MTL-1::mNG::3xFLAG expression in one genetic background. These data suggest many of the potential DAF-16 interactors we identified regulate stress responses.

**Fig. 1 | Proximity labeling of DAF-16 at normal and low DAF-2 insulin receptor signaling. a** Schematic showing knock-in of a TurboID::mNG::3xFLAG cassette just upstream of the *daf-16* stop codon. **b** Fluorescence images of the head region from gene-edited animals from (**a**) expressing DAF-16::TbID::mNG in a *daf-2(+)* (cytoDAF-16, left) or *daf-2(e1370)* (nucDAF-16, right) genetic background. 7 wild-type and 6 *daf-2(e1370)* animals were imaged with similar results. Scale bars: 50 μm. **c** Volcano plot showing log$_2$ fold changes in the levels of detected proteins in streptavidin purified extracts from cytoDAF-16 compared to WT (no TurboID expression). The bait protein, DAF-16, is marked with a red circle. Statistical analysis comparing experiments was performed using the Linear Models for Microarray Analysis (LIMMA, version 3.54.2) package in R. Moderated t-statistics were calculated using the limma-trend method with batch correction using the replicate number as batch,

and multiple testing correction was applied using the Benjamini-Hochberg (BH) method. **d** Volcano plot showing log$_2$ fold changes in the levels of detected proteins in streptavidin purified extracts from nucDAF-16 compared to cytoDAF-16. The identities of a subset of proteins, marked with a red circle, are indicated. Statistical analysis was as in (**c**). **e**, **f** Interactome plots of proteins enriched in nucDAF-16 vs cytoDAF-16 (**e**) or cytoDAF-16 vs nucDAF-16 (**f**). For panel (**e**) we used cut-off values of log$_2 \geq 3$ and $p \leq 0.05$; for (**f**) we used cut-off values of log$_2 \leq -2$ and $p \leq 0.05$. Contaminant proteins and proteins without human orthologs are not included. **g** Gene ontology (GO) analysis of enriched proteins in the nucDAF-16 vs cytoDAF-16 dataset. See also Supplementary Data 1, which provides an Excel file listing all data plotted in (**c**, **d**) with effect size and significance noted for each gene.

## ZFH-2 and DAF-16 form a complex, and ZFH-2 promotes longevity

Of the candidates predicted to be close to nuclear DAF-16, we further characterized ZFH-2, which belongs to a conserved family of TFs with multiple zinc finger and homeobox domains. We chose ZFH-2 because of the large (> 20-fold) increase in ZFH-2 labeling by DAF-16::TbID at low IIS (Fig. 2c and Supplementary Data 1), and since its mammalian ortholog ZFHX3 is a tumor suppressor[36], regulates nervous system development[37], and has been associated with obesity[38]. In *C. elegans*, null mutations in *zfh-2* cause lethality[39]. To visualize ZFH-2 expression and manipulate its function in time and space, we knocked in DNA encoding mScarlet(mSc)::AID::3xFLAG (AID, auxin-inducible degron) into the *zfh-2* locus, immediately upstream of the stop codon (Supplementary Fig. 2e). This should tag all predicted ZFH-2 isoforms, since they share their C-termini. ZFH-2::mSc::AID::3xFLAG was strongly expressed in most neurons, in sperm, and, to a lesser extent, in hypodermis and distal intestinal cells, and localized to the nucleus in all these cell types (Fig. 2d). To confirm our RNAi data, we analyzed MTL-1::mNG::3xFLAG expression in ZFH-2-depleted worms. AID-mediated knockdown[40] of ZFH-2 starting from the late L3 stage strongly induced MTL-1::mNG::3xFLAG expression (Fig. 2e, f). The much larger effect of AID compared to RNAi knockdown is consistent with RNAi by feeding working poorly in *C. elegans* neurons[41,42].

The proximity labeling data suggested ZFH-2 and DAF-16 could be part of a complex. To test this, we asked if they co-immunoprecipitated. We tagged DAF-16 by knocking in sequences encoding an mNG::HA tag just upstream of the *daf-16* stop codon. We grew *daf-2(e1370)* mutant animals expressing either *daf-16::mNG::HA* or *zfh-2::mSc::3xFLAG*, or both, and immunoprecipitated ZFH-2::mSc::3xFLAG. We then probed for co-IP of DAF-16::mNG::HA. DAF-16 and ZFH-2 consistently co-immunoprecipitated (Fig. 2g).

When DAF-2 receptor signaling is low, nuclear DAF-16 reprograms gene expression, and this substantially increases organismal lifespan[8,9]. To extend our biochemical data, we tested if ZFH-2 functionally interacts with DAF-2 and DAF-16 to regulate lifespan. Depleting ZFH-2 from early adulthood using AID substantially decreased the long lifespan of *daf-2(e1370)* mutants (Fig. 2h and Supplementary Table 1). By contrast, depleting ZFH-2 only marginally decreased lifespan in an otherwise WT background (Fig. 2i). ZFH-2 and DAF-16 likely act in the same pathway, since depleting ZFH-2 did not further decrease the lifespan either of *daf-16(mgDf50), daf-2(e1370)* or of *daf-16(mgDf50)* mutants (Fig. 2h, i).

DAF-2 and DAF-16 appear to regulate the expression of ZFH-2. Western blots showed that disrupting *daf-2* shifted the series of ZFH-2::mSc::3xFLAG bands that likely represent different splice isoforms to higher molecular weights. This upshift was reversed when the *daf-2* mutants lacked *daf-16* (Supplementary Fig. 2f). A simple interpretation is that changes in IIS alter the levels of specific ZFH-2 isoforms in a DAF-16-dependent way.

Many genes regulated by DAF-16 have been identified[8]. Three such genes are *mtl-1*, the superoxide dismutase *sod-3*, and the alcohol

dehydrogenase *adh-1* (previously called *dod-11*). We performed RT-qPCR analysis to probe if ZFH-2 regulates these genes. ZFH-2::AID knockdown reduced mRNA levels for *sod-3* but not for *mtl-1* or *adh-1* (Supplementary Fig. 2g). Together, our data suggest that ZFH-2 and DAF-16 form a complex and cooperate to promote longevity when DAF-2 signaling is low, but further experiments are needed to establish how ZFH-2 regulates lifespan.

## DAF-2 promotes formation of a PAR-1–DAF-16 complex that inhibits DAF-16 and shortens lifespan

DAF-16 subcellular localization is regulated by phosphorylation, notably by the AKT-1 and AKT-2 kinases, whose activation by DAF-2 receptor signaling promotes cytosolic localization of DAF-16[31,32]. Our proximity labeling experiments detected AKT-1 and AKT-2 but did not highlight a DAF-2-regulated interaction between them and DAF-16 (Supplementary Data 1, 2). By contrast, proximity labeling suggested DAF-2 signaling increased DAF-16's interaction with the PAR-1 (partitioning defective gene 1) kinase 8-fold (Fig. 3a).

PAR-1, and its mammalian orthologs, the microtubule affinity regulating kinases (MARK), play critical roles in establishing animal cell polarization and regulating cytoskeletal dynamics[43]. MARK kinases are expressed broadly in mammals, and mouse mutants defective in MARK kinases have a range of phenotypes, including reduced adiposity, defective gluconeogenesis, and altered growth, fertility, metabolism, and learning and memory[44]. Our TurboID data led us to speculate that IIS regulates PAR-1/MARK kinase signaling, and that in turn, PAR-1/MARK kinase regulates DAF-16/FOXO function. We decided to test this hypothesis, since if correct, it would link two intensively studied signaling pathways. Moreover, since PAR-1 is a kinase, its potential physical interaction with DAF-16 would predict it regulates DAF-16 via phosphorylation, which is readily testable.

In *C. elegans*, PAR-1 is required to establish cell asymmetries during early embryonic divisions, and *par-1* null mutations confer lethality[45]. We therefore knocked down PAR-1 post-embryonically using AID. To target all 13 PAR-1 isoforms, we used gene-editing to insert sequences encoding mSc::AID::3xFLAG in frame into the 11th *par-1* exon (between A895 and A896 in isoform a), which is common to all isoforms (Supplementary Fig. 3a). Edited animals showed no overt phenotype, and expressed PAR-1::mSc::AID::3xFLAG in hypodermis and neurons, with notable enrichment in the nerve ring, an area of dense neuropil and extensive synaptic contacts (Fig. 3b and Supplementary Fig. 3b). AID-mediated somatic depletion of PAR-1 from hatching caused a small developmental delay and resulted in adult animals having a protruding vulva (P-vul phenotype). To avoid this, we depleted PAR-1 starting from later stages. Depleting PAR-1 after the late L3 stage significantly increased MTL-1::mNG::3xFLAG expression, consistent with our RNAi data (Fig. 3c, d).

To investigate if DAF-16 and PAR-1 form a complex, we performed co-IP experiments using our epitope-tagged alleles. Although kinases and their protein substrates often interact transiently, we could consistently co-IP DAF-16::mNG::HA and PAR-1::mSc::FLAG (Fig. 3e). Thus,

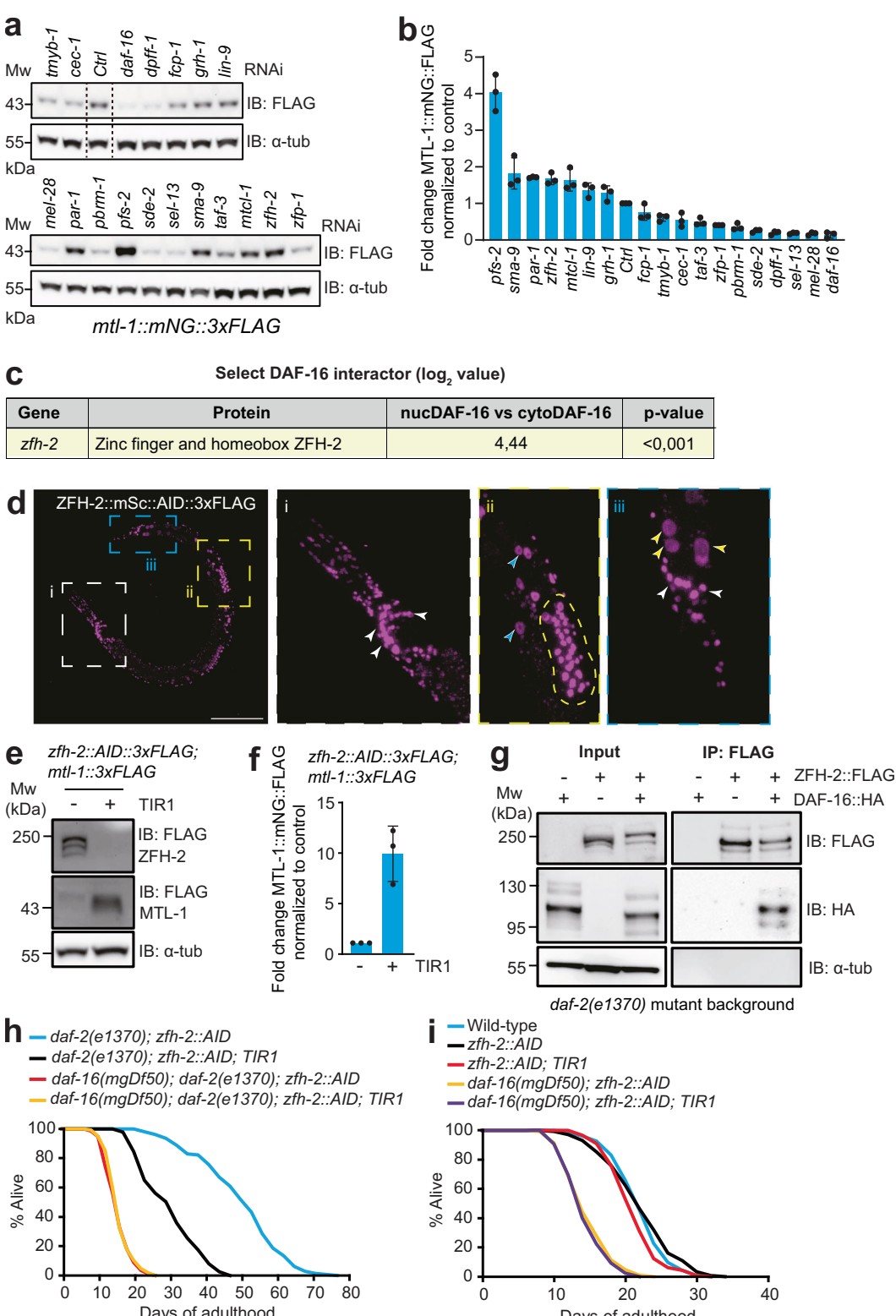

in vivo proximity labeling and ex vivo co-IP suggest DAF-16 and PAR-1 strongly interact and are in a complex.

To test if DAF-16 activity is regulated by PAR-1, we first compared the subcellular localization of DAF-16::mNG::HA in different genetic backgrounds: *daf-2(e1370)*, *akt-1(ok525)* and a strain grown on auxin that expresses the *par-1::mSc::AID::3xFLAG* knockin together with ubiquitous somatic expression of TIR1. PAR-1 knockdown significantly

increased nuclear localization of DAF-16::mNG::HA (Fig. 3f, g; see Supplementary Fig. 3c for scoring of DAF-16 localization). These data suggest that PAR-1, like AKT-1 and AKT-2, promotes cytoplasmic localization of DAF-16. Additionally, Western blot analyses revealed that PAR-1 knockdown increased DAF-16 protein levels twofold compared to controls (Fig. 3h, i), but did not alter *daf-16* mRNA levels (Supplementary Fig. 3d). These data suggest PAR-1 destabilizes DAF-16.

**Fig. 2 | The obesity-related zinc finger and homeobox transcription factor ZFH-2 forms a complex with DAF-16 and has pro-longevity functions. a** Western blot showing levels of FLAG-tagged MTL-1 following RNAi knockdown of potential DAF-16 interactors identified by TurboID. An empty vector (Ctrl) and *daf-16* RNAi are included as controls. RNAi knockdown is in a WT genetic background. α-tubulin provides a loading control. **b** MTL-1::FLAG band intensity (see panel **a**) is normalized first to the α-tubulin loading control and then to the empty vector RNAi control (*n* = 3 independent biological repeats). **c** DAF-16::TurboID proximity labeling detects ZFH-2 20-fold more strongly in a *daf-2(e1370)* mutant (nucDAF-16) than a wild-type (cytoDAF-16) genetic background. See Supplementary Data 1. **d** Confocal images of a transgenic animal expressing gene edited ZFH-2::mSc::AID::3xFLAG. ZFH-2 is expressed in many neurons, in the germline, and weakly in the hypodermis and posterior gut. White arrowheads indicate neuronal, blue arrowheads hypodermal, and yellow arrowheads distal gut cell nuclei. The yellow dashed line outlines the spermatheca. 9 animals were imaged with similar results. Scale bar: 100 μm. **e** AID-mediated depletion of ZFH-2 increases expression of FLAG-tagged MTL-1 as assayed by Western blot. α-tubulin provides a loading control. **f** MTL-1::FLAG band intensity (see panel **e**) is normalized to the α-tubulin loading control (*n* = 3 independent biological repeats). Animals were grown on NGM plates to the late L3 stage and transferred to auxin-containing NGM plates for 12 h before harvesting for Western blot analysis. **g** HA-tagged DAF-16 forms a complex with FLAG-tagged ZFH-2. Experiments were conducted in a *daf-2(e1370)* mutant background. DAF-16::HA consistently co-immunoprecipitates with ZFH-2::FLAG (*n* = 3 independent biological repeats). **h, i** Knockdown of ZFH-2 using AID shortens the lifespan of *daf-2* mutants (**h**), but not of wild-type animals, *daf-16* mutants (**i**), or *daf-16; daf-2* double mutants (**h**). Knockdown was initiated by transferring animals to auxin plates at the L4/young adult stage. TIR1 was expressed ubiquitously in the soma. See Supplementary Table 1 for values and statistical analyses of lifespan assays. Source data are provided as a Source Data file.

To test the physiological effects of PAR-1 depletion on DAF-16 function, we measured organismal lifespan. A previous study showed that RNAi knockdown of *par-1* lengthened lifespan, but suggested that *daf-16* and *daf-2* were not involved in this regulation[46]. We selectively depleted PAR-1 from late L4/early adulthood by transferring animals to auxin plates at this time. As a control, we measured the lifespan of animals kept on auxin but lacking the TIR1 co-factor required for AID-mediated degradation. Depleting PAR-1 significantly increased *C. elegans* lifespan (Fig. 3j). This extended lifespan phenotype was fully dependent on DAF-16: depleting PAR-1 did not alter the lifespan of *daf-16* deletion mutants, consistent with PAR-1 and DAF-16 acting in the same pathway (Fig. 3j). Moreover, depleting PAR-1 did not further increase the lifespan of *daf-2(e1370)* mutants (Fig. 3k). Together, these data suggest that PAR-1 participates in IIS and exerts its effects on longevity by regulating DAF-16, including inhibiting DAF-16 nuclear localization and protein levels.

## Pharmacological extension of lifespan by a MARK / PAR-1 kinase inhibitor

PAR-1 exerts its effects predominantly via its kinase activity. To extend our PAR-1–AID-knockdown analysis, we asked if pharmacologically inhibiting PAR-1 kinase activity prolonged organismal lifespan in a DAF-16-dependent manner. Compound 39621 is a highly selective inhibitor of MARK family kinases[47]. It acts by competing with ATP for binding the kinase active site, with an IC50 of 3.6 μM at an ATP concentration of 100 μM. Since 39621 is cell-permeable, we expected *C. elegans* tissues to take it up from the medium. We transferred wild-type and *daf-16(mgDf50)* null mutants at the L4/young adult stage to NGM plates containing 20 μM compound 39621, and kept them on the compound while we performed lifespan assays. This concentration of 39621 strongly inhibits MARK2 in tissue culture cells[47]. Like PAR-1-AID knockdown, this treatment significantly extended *C. elegans* lifespan in a DAF-16-dependent way (Fig. 3l).

## DAF-2 and PAR-1 promote DAF-16 phosphorylation at a conserved S249 residue

Our co-IP data suggested that PAR-1 and DAF-16 form a complex. Our genetic epistasis studies placed PAR-1 in the DAF-2 insulin receptor signaling pathway, regulating DAF-16 activity and lifespan. Our pharmacology suggested that PAR-1's effects on longevity required its kinase activity and were DAF-16-dependent. We therefore examined whether disrupting PAR-1 altered DAF-16 phosphorylation. For comparison, we also included *daf-2* and *akt-1* mutants in our proteomic analysis.

We grew animals that expressed the CRISPR-edited *daf-16::mNG::HA* gene in four genetic backgrounds: wild-type, *daf-2(e1370)*, *akt-1(ok525)*, and *par-1::mSc::AID::3xFLAG; eft-3p::TIR1::tagBFP* (on auxin-supplemented plates). For all genotypes, animals were grown in well-fed and uncrowded conditions at a room temperature of 21 °C; we observed no significant dauer formation under these conditions in any strain. We affinity purified DAF-16::mNG::HA from each strain and looked for changes in DAF-16 phosphorylation by mass spectrometry. DAF-16 was strongly and consistently hypophosphorylated at residues T102, S249, T273, S345 and S348 (notation refers to isoform h) in *daf-2(e1370)* mutants compared to wild-type animals (Fig. 4a and Supplementary Data 3). We observed a similar strong loss of phosphorylation at S249 in PAR-1 depleted worms (Fig. 4a and Supplementary Data 3). In *akt-1(ok525)* mutants, we observed reduced phosphorylation of DAF-16 at residues T102 and S179 but not S249 (Fig. 4a and Supplementary Data 3; see Supplementary Data 3 for the MS2 spectra of all phosphosites identified). Residue S249 is located towards the end of the FOXO DNA binding domain of DAF-16, in a stretch of residues highly conserved among FOXO proteins encoded by the worm, fly, mouse and human genomes (Fig. 4b). Bioinformatic analysis predicts a PAR-1/MARK phosphorylation site at S249 (www.phosphosite.org/kinaseLibraryAction).

## A phosphomimetic S249D mutation inhibits DAF-16 function and promotes its cytoplasmic localization

To investigate whether phosphorylation at S249 alters DAF-16 subcellular location and physiological function, we further edited the *daf-16::mNG::HA* gene to change codon 249 to encode either a non-phosphorylatable alanine (S249A) or a phosphomimetic aspartate (S249D). We then compared the subcellular localization of DAF-16(WT)::mNG, DAF-16(S249A)::mNG, and DAF-16(S249D)::mNG proteins in wild-type and *daf-2(e1370)* backgrounds using well-fed, unstressed animals. In a wild-type background, DAF-16 S249A::mNG showed significantly higher nuclear localization than DAF-16(WT)::mNG and DAF-16(S249D)::mNG, which were both excluded from the nucleus (Fig. 4c, d). The subcellular localization of the different DAF-16 variants in a *daf-2(e1370)* genetic background was age-dependent. In larvae and young adults, all three proteins localized to the nucleus. However, in Day 4 and Day 10 adults, DAF-16(S249A)::mNG was more localized to the nucleus than DAF-16(WT)::mNG, whereas DAF-16(S249D)::mNG was less nuclear (Fig. 4e and Supplementary Fig. 4a). These data are consistent with our findings that DAF-16::mNG accumulates in the nucleus in PAR-1-AID knockdown animals, and suggest that phosphorylation of S249 promotes cytoplasmic localization of DAF-16.

To test how S249 phosphorylation alters DAF-16 function, we first studied dauer formation. Dauers are a stress-resistant alternative larval stage formed by *C. elegans* when conditions are unfavorable for reproductive growth[48]. The developmental decision to form a dauer is regulated by food availability, population density, and temperature. Wild-type animals grown at low population density with plentiful food at 25 °C form few dauers. By contrast, mutants defective in IIS, for example, *daf-2(e1370)* animals, form close to 100% dauers under these conditions[48]. Dauer formation in *daf-2(e1370)* animals depends on functional DAF-16: *daf-2(e1370); daf-16(null)* animals do not form

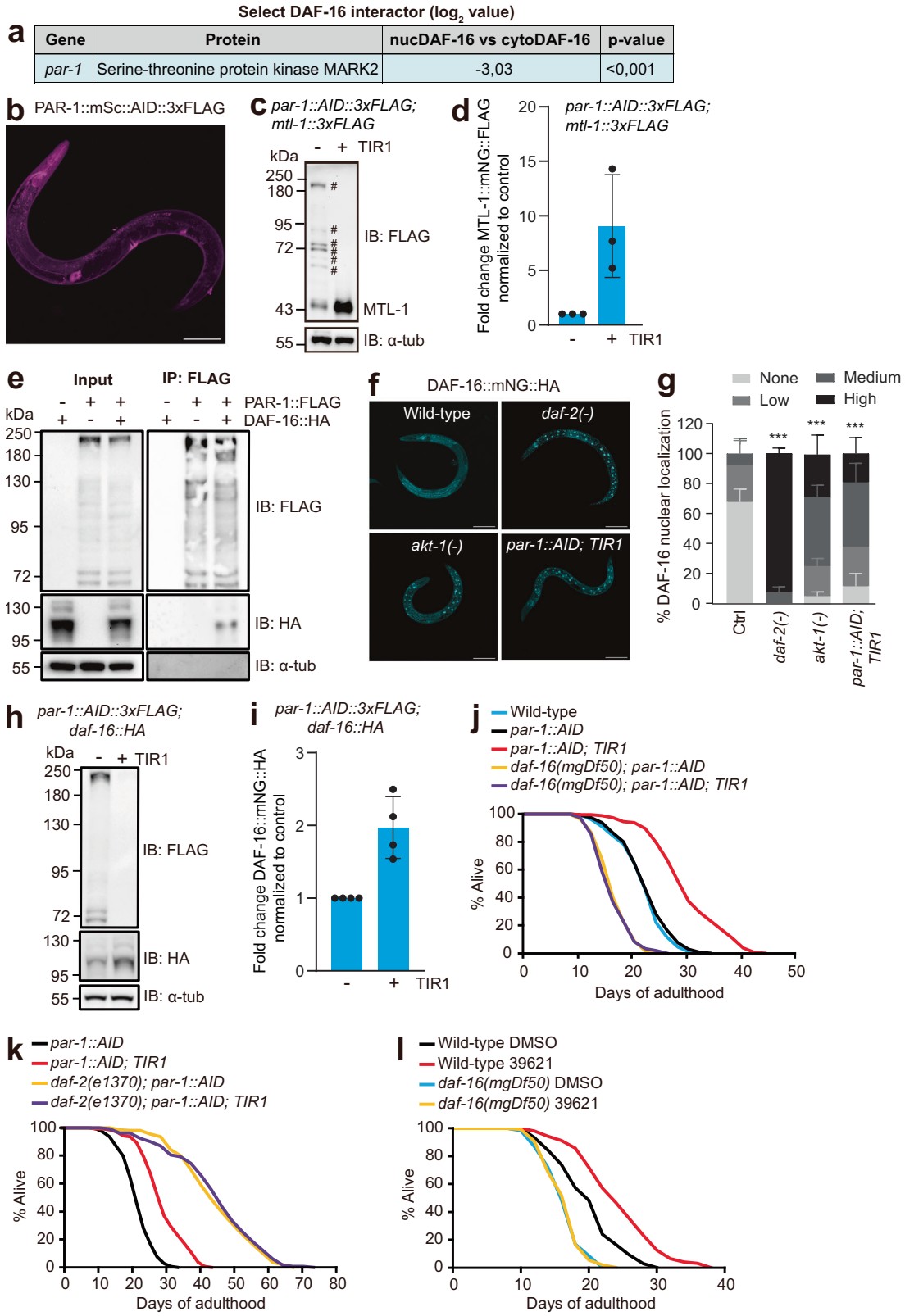

dauers[48]. The *daf-16(S249D)* allele almost fully suppressed the constitutive dauer formation phenotype of *daf-2(e1370)* mutants, whereas the *daf-16(S249A)* allele had no detectable effect (Fig. 4f). Thus, phosphomimetic DAF-16 S249D is unable to induce dauer formation when DAF-2 signaling is defective, despite localizing to the nucleus in larvae.

Disrupting *daf-2* also reduces brood size[49]. In a *daf-2(+)* background, both *daf-16(S249D)* and *daf-16(S249A)* animals had a normal brood size (Supplementary Fig. 4b). However, the *daf-16(S249D)* allele, but not *daf-16(S249A)*, restored a normal brood size to *daf-2(e1370)* mutants (Supplementary Fig. 4b), again

**Fig. 3 | PAR-1 kinase forms a complex with DAF-16 and inhibits its activity.**
**a** Proximity labeling by DAF-16::TurboID detects PAR-1 8-fold more strongly in *daf-2(+)* animals (cytoDAF-16) compared to *daf-2(e1370)* mutants (nucDAF-16). See Supplementary Data 1. **b** Confocal image of an animal expressing PAR-1::mScarlet::AID::3xFLAG from the CRISPR-edited *par-1* locus. 8 animals were imaged with similar results. Scale bar: 100 μm. **c** AID-mediated knockdown of PAR-1 increases expression of FLAG-tagged MTL-1 as assayed by Western blot. Protein bands marked with # correspond to different PAR-1 isoforms. α-tubulin provides a loading control. **d** MTL-1::FLAG band intensity (see panel **c**) normalized to the α-tubulin loading control. Animals were grown on NGM plates until the late L3 stage and then transferred to auxin-containing NGM plates for 12 h prior to harvesting for Western blot analysis (*n* = 3 independent biological repeats). **e** HA-tagged DAF-16 is in a complex with FLAG-tagged PAR-1. Immunoprecipitating PAR-1::FLAG consistently pulls down DAF-16::HA (n = 3 independent biological repeats). **f** Confocal images of DAF-16::mNG::HA expressed from the edited *daf-16* gene in wild-type (WT), *daf-2(e1370)*, *akt-1(ok525)* and *par-1::mSc::AID; TIR1* (+ auxin) backgrounds. DAF-16 is cytoplasmic in wild-type animals but nuclear in the mutants or AID knockdown. Worm were imaged at the L4 stage. Scale bars: 100 μm. **g**, Quantification of DAF-16 nuclear localization (see **f**) in wild-type (Ctrl), *daf-2(e1370)*, *akt-1(ok525)* and *par-1::mSc::AID; TIR1* (+ auxin) backgrounds at L4 stage (*n* = 3 independent biological repeats, total > 50 animals were scored per condition). Error bars represent s.e.m; Statistical comparisons are to the control; \*\*\**p* < 0.001. *p*-values were calculated using the chi-squared test. **h** AID-mediated knockdown of PAR-1 increases levels of HA-tagged DAF-16 as assayed by Western blot. α-tubulin provides a loading control. **i** DAF-16::HA band intensity (see panel **h**) is normalized to the α-tubulin loading control. Animals were grown on NGM plates until the early L3 stage and then transferred to auxin-containing NGM plates overnight prior to harvesting for Western blot analysis (*n* = 4 independent biological repeats). **j** Depleting PAR-1 using AID from young adulthood significantly increases the lifespan of *daf-2(+)* animals. This lifespan extension was fully dependent on *daf-16*. **k** Depleting PAR-1 as in (**j**) did not further extend the lifespan of *daf-2(e1370)* animals. The *par-1::AID* and *par-1::AID; TIR1* data is the same as in (**j**). See Supplementary Table 2 for values and statistical analyses of lifespan assays. **l** Inhibiting PAR-1 kinase activity from young adulthood using the MARK2 inhibitor 39621 significantly increases the lifespan of wild-type animals. This lifespan extension was fully dependent on *daf-16*. See Supplementary Table 3 for values and statistical analyses of lifespan assays. Source data are provided as a Source Data file.

suggesting that the phosphomimetic S249D substitution reduces DAF-16 activity.

We next examined lifespan. Compared to control animals expressing wild-type DAF-16::mNG, animals expressing DAF-16(S249A)::mNG exhibited a modest but significant increase in lifespan (Fig. 4g and Supplementary Fig. 4c). By contrast, animals that expressed DAF-16(S249D)::mNG showed a decreased lifespan compared to the control worms (Fig. 4g and Supplementary Fig. 4c). We also examined lifespan in a *daf-2(e1370)* mutant background. As expected, *daf-2(e1370); daf-16::mNG* animals lived twice as long as *daf-2(+)* controls (Fig. 4h). The *daf-16(S249A)::mNG* allele did not alter this long lifespan, but the *daf-16(S249D)::mNG* allele shortened it substantially (Fig. 4h and Supplementary Fig. 4d). These data suggest that the S249A substitution partially uncouples DAF-16 activity from negative regulation by DAF-2, whereas the phosphomimetic S249D substitution leads to constitutively inhibited DAF-16, even when DAF-2 is defective. These data are consistent with a model in which phosphorylation of S249 by PAR-1 in well-fed, unstressed animals inhibits DAF-16 activity.

### DAF-16 S249D prevents PAR-1 knockdown from extending lifespan

If PAR-1 inhibits DAF-16 predominantly via phosphorylation of S249, then knocking down *par-1* should not change the longevity of animals expressing DAF-16(S249D). To test this, we compared the lifespans of animals expressing DAF-16(WT)::mNG or DAF-16(S249D)::mNG in a *daf-2(+)* or *daf-2(e1370)* background when we depleted PAR-1. In a *daf-2(+)* background, PAR-1 depletion increased the lifespan of *daf-16(WT)* but not of *daf-16(S249D)* mutant worms (Fig. 5a and Supplementary Fig. 5a). In addition, PAR-1 depletion did not extend the lifespans of *daf-2(e1370); daf-16(WT)* or *daf-2(e1370); daf-16(S249D)* worms (Fig. 5b and Supplementary Fig. 5b). These data are consistent with the effects of PAR-1 on lifespan being mediated via phosphorylation of DAF-16 S249, and support a model in which PAR-1 is normally activated by IIS to regulate lifespan. Not all effects of PAR-1 knockdown are, however, mediated via phosphorylation of S249. Whereas AID knockdown of PAR-1 caused a 2-fold increase in DAF-16(WT)::mNG protein levels (Fig. 3h, i), the levels of the non-phosphorylatable DAF-16(S249A)::mNG were only marginally elevated compared to DAF-16(WT)::mNG, and DAF-16(S249D)::mNG levels were unaltered (Supplementary Fig. 5c, d). Moreover, knockdown of PAR-1 by AID increased the levels of DAF-16 S249A, DAF-16 S249D and DAF-16(WT) proteins similarly (Supplementary Fig. 5e, f), suggesting PAR-1 regulates DAF-16 protein levels independently of S249 phosphorylation state. These data may help explain why the lifespan extension

conferred by DAF-16 S249A appears smaller than that conferred by PAR-1 knockdown (compare data in Fig. 3j and Supplementary Fig. 4c with Fig. 4g and Supplementary Fig. 5a).

In canonical IIS, AKT-1 and AKT-2 phosphorylate DAF-16, leading to its exclusion from the nucleus and inhibiting its function. To dissect the relationship between the regulation of DAF-16 by AKT-1/AKT-2 on the one hand, and by PAR-1 on the other, we analyzed the lifespan of *akt-1; akt-2* double mutants that express wild-type *daf-16* or *daf-16(S249D)*. As reported previously[24], loss of *akt-1* and *akt-2* substantially increased the lifespan of worms expressing wild-type DAF-16 (Fig. 5c and Supplementary Fig. 5g). Disrupting *akt-1* and *akt-2* also extended the lifespan of animals expressing DAF-16(S249D) but to a much lesser extent (Fig. 5d and Supplementary Fig. 5h). This contrasts with our results with PAR-1 knockdown, where the lifespan extension conferred by PAR-1 knockdown was completely suppressed by the *daf-16(S249D)* allele (Fig. 5a and Supplementary Fig. 5a). Thus, AKT-1 and AKT-2 can regulate lifespan independently of DAF-16 phosphorylation at S249.

### Regulation of PAR-1
Extensive studies of cell polarization in the early embryo have delineated intricate regulation of PAR-1 activity[43,50]. One critical PAR-1 regulator is the atypical protein kinase C, PKC-3, which phosphorylates PAR-1 at threonine 983 and thereby inhibits PAR-1 kinase activity[51,52]. This inhibition is blocked in a PAR-1 T983A mutant. To probe if PKC-3 regulates PAR-1 to impact lifespan, we studied animals expressing a *par-1(T983A)* allele. These animals showed a shortened lifespan, consistent with increased PAR-1 T983A activity reducing DAF-16 function (Fig. 5e).

A second mechanism that regulates PAR-1 activity involves the C-terminal KA1 domain[50,53,54]. KA1 contains a basic surface that binds negatively charged phospholipids and targets PAR-1 to membranes. The same basic amino acids inhibit PAR-1 kinase activity, and phospholipid binding relieves this inhibition. Deleting the KA1 domain (PAR-1 ΔKA1), or disrupting the KA1 basic surface (PAR-1 K1170S R1171S), increases basal PAR-1 kinase activity in vitro[50]. To investigate whether the PAR-1 KA1 domain influences DAF-16 activity and longevity, we asked if *par-1(ΔKA1)* or *par-1(KRSS)* mutations altered lifespan. Both alleles reduced lifespan (Fig. 5f). Together, these data are consistent with a model in which inhibition of PAR-1 kinase activity, partly by PKC-3, impacts lifespan. Further work is required to identify pathways that regulate lifespan by modulating PKC-3.

### PAR-1 and MARK2 phosphorylate DAF-16 in vitro
While our mass spectrometry analyses suggested that PAR-1 knockdown reduced phosphorylation at DAF-16 S249, it did not address

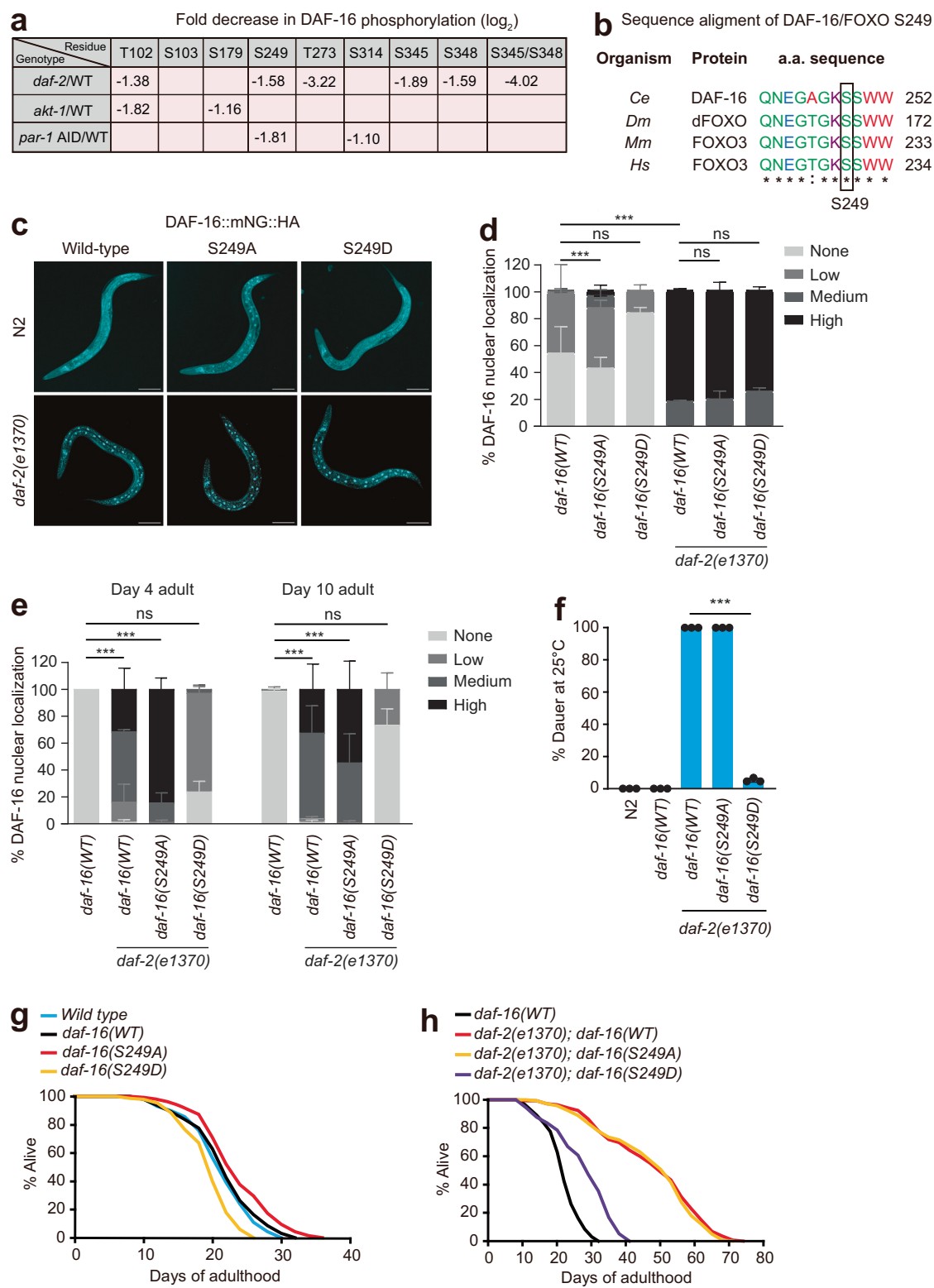

**a** Fold decrease in DAF-16 phosphorylation (log$_2$)

| Genotype \ Residue | T102 | S103 | S179 | S249 | T273 | S314 | S345 | S348 | S345/S348 |
|---|---|---|---|---|---|---|---|---|---|
| *daf-2*/WT | -1.38 | | | -1.58 | -3.22 | | -1.89 | -1.59 | -4.02 |
| *akt-1*/WT | -1.82 | | -1.16 | | | | | | |
| *par-1* AID/WT | | | | -1.81 | | -1.10 | | | |

**b** Sequence aligment of DAF-16/FOXO S249

| Organism | Protein | a.a. sequence |
|---|---|---|
| *Ce* | DAF-16 | QNEGAGKSSWW 252 |
| *Dm* | dFOXO | QNEGTGKSSWW 172 |
| *Mm* | FOXO3 | QNEGTGKSSWW 233 |
| *Hs* | FOXO3 | QNEGTGKSSWW 234 |

S249

**c** DAF-16::mNG::HA

Wild-type   S249A   S249D
N2
*daf-2(e1370)*

**d** % DAF-16 nuclear localization
None / Low / Medium / High
*daf-16(WT)* / *daf-16(S249A)* / *daf-16(S249D)* / *daf-2(e1370)*

**e** Day 4 adult / Day 10 adult
% DAF-16 nuclear localization
None / Low / Medium / High
*daf-16(WT)* / *daf-16(S249A)* / *daf-16(S249D)* / *daf-2(e1370)*

**f** % Dauer at 25°C
N2 / *daf-16(WT)* / *daf-16(WT)* / *daf-16(S249A)* / *daf-16(S249D)* / *daf-2(e1370)*

**g** — *Wild type* — *daf-16(WT)* — *daf-16(S249A)* — *daf-16(S249D)*
% Alive / Days of adulthood

**h** — *daf-16(WT)* — *daf-2(e1370); daf-16(WT)* — *daf-2(e1370); daf-16(S249A)* — *daf-2(e1370); daf-16(S249D)*
% Alive / Days of adulthood

whether the effect was direct or indirect. To probe this, we performed in vitro kinase assays using recombinant PAR-1 (aa 1-482; T325E kinase active form ref. 55; Supplementary Fig. 6a) and human MARK2 kinases and synthetic peptides that correspond to sequences from DAF-16 and FOXO3a surrounding the S249 residue as substrates. As a positive control, we showed that both PAR-1 and MARK2 efficiently phosphorylated a purified MEX-5 fragment (Fig. 6a–c and Supplementary

Fig. 6b), an RNA-binding protein known to be directly phosphorylated by PAR-1[55]. MARK2 phosphorylated both the FOXO3a and S249WT peptide (Fig. 6a, b). Phosphorylation was significantly reduced when we used a synthetic peptide with an S249A substitution, and was at baseline levels with a peptide harboring S249A S250A substitutions, suggesting that MARK2 phosphorylates S249, and to a lesser extent S250. Basal level of phosphorylation in the absence of a

**Fig. 4 | PAR-1 promotes DAF-16 phosphorylation and its accumulation in cytoplasm. a** Fold decrease in DAF-16 phosphorylation at indicated residues in *daf-2(e1370)*, *akt-1(ok525)* and *par-1::mSc::AID* genetic backgrounds compared to wild-type (WT) controls. See also Supplementary Data 3. Residue numbers refer to DAF-16 isoform h (www.wormbase.org). **b** Alignment of the DAF-16 S249 phosphorylation site with homologous sequences from FOXO orthologs. *Ce Caenorhabditis elegans; Dm Drosophila melanogaster; Mm Mus musculus; Hs Homo sapiens.* * completely conserved;: conserved in ¾ sequences. **c** Confocal images of L4 animals showing localization of the indicated DAF-16 variants, expressed from the gene-edited endogenous *daf-16* locus, in an N2 (wild-type) or *daf-2(e1370)* genetic background. **d, e** Nuclear localization of the indicated DAF-16 variants quantified in an N2 (wild-type) or *daf-2(e1370)* genetic background. **d** Late L4/Young adults (*n* = 3 independent biological repeats, > 65 animals were scored per condition); **e**, 4- and 10- day old adults (*n* = 3 independent biological repeats, > 70 animals were scored per condition). Error bars represent s.e.m; ns, *p > 0.05; ***p < 0.001, chi-squared test. **f** Dauer formation of wild-type (N2), *daf-16(WT)::mNG, daf-2(e1370); daf-16(WT)::mNG, daf-2(e1370); daf-16(S249A)::mNG* or *daf-2(e1370); daf-16(S249D)::mNG* gene-edited animals. *n* = 3 biological repeats and ≥ 1500 animals for each condition. Error bars represent s.e.m; ***p < 0.001, two-tailed Student's *t* test. **g** Lifespan analysis of wild-type (N2), *daf-16(WT)::mNG, daf-16(S249A)::mNG* or *daf-16(S249D)::mNG* gene-edited animals. **h** Lifespan analysis of *daf-16(WT)::mNG, daf-2(e1370); daf-16(WT)::mNG, daf-2(e1370); daf-16(S249A)::mNG* and *daf-2(e1370); daf-16(S249D)::mNG* transgenic animals. The *daf-16(WT)::mNG* data are the same as in (**g**). See Supplementary Table 4 for values and statistical analyses of lifespan assays.

peptide substrate suggested autophosphorylation by MARK2 (Fig. 6b). Although less active than MARK2, PAR-1 also phosphorylated FOXO3a and S249WT peptides (Fig. 6c). Phosphorylation was also dependent on the S249 and S250 residues (Fig. 6c). Autophosphorylation of PAR-1 was higher than for MARK2 (Fig. 6b, c). Together, our data suggest that PAR-1 can phosphorylate the DAF-16 S249 residue, and that MARK kinases may similarly regulate human FOXOs.

## Discussion

Using proximity labeling, we identified proteins that preferentially interact with the pro-longevity transcription factor DAF-16/FOXO either when DAF-2 insulin receptor signaling is active or when it is disrupted. By studying two of these potential interactors, we highlight pathways previously not known to regulate *C. elegans* longevity.

Canonically, DAF-2 insulin receptor signaling negatively regulates DAF-16 primarily by activating the AKT-1 and AKT-2 kinases. These kinases phosphorylate DAF-16 and lead to its exclusion from the nucleus. We show here that in well-fed, unstressed animals, the PAR-1/MARK kinase, extensively studied in the context of cell polarity, provides additional, functionally important inhibition of DAF-16 downstream of DAF-2 receptor signaling. DAF-2 signaling promotes the formation of a PAR-1/DAF-16 complex and stimulates PAR-1-dependent phosphorylation of DAF-16 S249. S249 lies in the Forkhead DNA binding domain, in a sequence highly conserved across FOXO transcription factors. Knocking down PAR-1 promotes DAF-16 nuclear entry and extends *C. elegans* lifespan in a DAF-16-dependent manner, but does not extend the lifespan of *daf-2* mutants; it also doubles DAF-16 protein levels independently of S249 phosphorylation or altered *daf-16* mRNA levels. These data are consistent with PAR-1 inhibiting DAF-16 in response to DAF-2 signaling to regulate lifespan.

Studies of gene-edited animals expressing phosphomimetic DAF16 S249D, or phosphorylation-incompetent DAF-16 S249A, suggest that an important way PAR-1 inhibits DAF-16 function is by promoting S249 phosphorylation. DAF-16 S249D reduces the lifespan of both wild-type and long-lived *daf-2* mutants. It also suppresses the dauer-constitutive phenotype and reduced brood size of *daf-2* mutants, and promotes cytoplasmic localization of DAF-16. Conversely, DAF-16 S249A shows increased nuclear localization and prolongs the lifespan of otherwise wild-type animals, although not to the same extent as PAR-1 knockdown. Recombinant human MARK2 and *C. elegans* PAR-1 can phosphorylate FOXO3a- and DAF-16-derived peptides harboring S249 in vitro, suggesting that DAF-16 is directly phosphorylated by PAR-1 in vivo, and that MARK kinases phosphorylate FOXO transcription factors in humans.

Disrupting the DAF-2 receptor, like knocking down PAR-1, reduces DAF-16 phosphorylation at S249. This suggests that DAF-2 signaling promotes PAR-1 kinase activity. A study that analyzed changes in the phosphoproteome of *C. elegans* in insulin/IGF-1 pathway mutants[56] supports this conclusion: predicted PAR-1 substrates showed reduced phosphorylation in *daf-2(e1370)* animals, consistent with generally reduced PAR-1 activity[56].

How does DAF-2 signaling activate PAR-1? Studies of cell polarity have delineated a network of interactions that regulate PAR-1/MARK kinases[43,50]. In the early *C. elegans* embryo, the atypical protein kinase PKC-3 phosphorylates PAR-1 at T983 to inhibit PAR-1 kinase activity, impede its localization at the plasma membrane, and enhance its binding to 14-3-3 adapter proteins. In addition, the C-terminal KA1 domain of PAR-1 is autoinhibitory, but inhibition can be relieved by phospholipid binding[50]. Both these regulatory mechanisms appear to impact aging. Animals expressing PAR-1(T983A) or PAR-1(ΔKA1) mutant proteins have shorter lifespans than controls, consistent with the associated disinhibition of PAR-1 activity reducing DAF-16 function. Further work is required to elucidate how DAF-2 signaling regulates PAR-1. In other contexts, insulin receptor signaling activates aPKC by stimulating PDK; this would be expected to repress, not activate, PAR-1[57].

Nuclear-to-cytosol shuttling of DAF-16 is regulated by several kinases including AKT-1, AKT-2, SGK-1, AMPK, CST-1 and JNK-1[25,58–61]. All of these kinases, except JNK-1, were detected by our DAF-16 proximity labeling. However, their labeling by DAF-16::TurboID was unaltered when DAF-2 signaling was disrupted. Perhaps these kinases interact more transiently with DAF-16 than PAR-1. Previous work has predicted multiple phosphorylation sites in DAF-16[56,59]. A subset of these predicted sites, T54, S271, T273 and S348, are predicted Akt phosphorylation sites[59]. Simultaneously mutating all of these residues to alanine causes DAF-16 to accumulate in the nucleus, but does not extend lifespan in wild-type worms[59]. Thus, the AKT-1/2 phosphorylation site/s on DAF-16 relevant for influencing lifespan are not known.

A previous study reported that RNAi knockdown of PAR-1 extends *C. elegans* lifespan but suggested PAR-1 exerts its effects independently of insulin signaling and DAF-16[46]. They suggested instead that PAR-1 acts through the nutrient-responsive S6 kinase and the AMP-activated protein kinase alpha-subunit (AMPKa) AAK-2 to regulate lifespan. In particular, they reported that PAR-1 knockdown activates AAK-2 in older animals. Increased AAK-2 activity is known to increase *C. elegans* lifespan by both DAF-16-dependent and independent mechanisms[62]. Further work is required to understand how PAR-1 inhibits AAK-2.

DAF-16 proximity labeling in animals with low DAF-2 activity, when DAF-16 is nuclear, highlighted multiple transcription regulators and chromatin modifiers. A few of these were previously implicated in regulating either DAF-16 activity, such as the chromatin remodeller SWI/SNF (PBRM-1; SWSN-3; PHD-10)[10] or stress responses, such as the double PHD finger transcription factor *DPFF-1*[63], and Grainyhead1 (GRH-1), which is required for *daf-2* lifespan extension and has an evolutionarily conserved role in IIS[33]. Other potential interactors mediate responses known to be regulated by DAF-16, but have not previously been shown to interact biochemically with DAF-16. For example, DAF-16 helps mediate DNA damage responses[64], and we identify the DNA damage recognition and repair factors XPC-1 (Xeroderma pigmentosum group C), RFC-4 (replication factor C subunit 4), and SEL-13 (ortholog of human ZNF830) as prominent DAF-16 interactors at low DAF-2 activity. We speculate DAF-16 directly participates

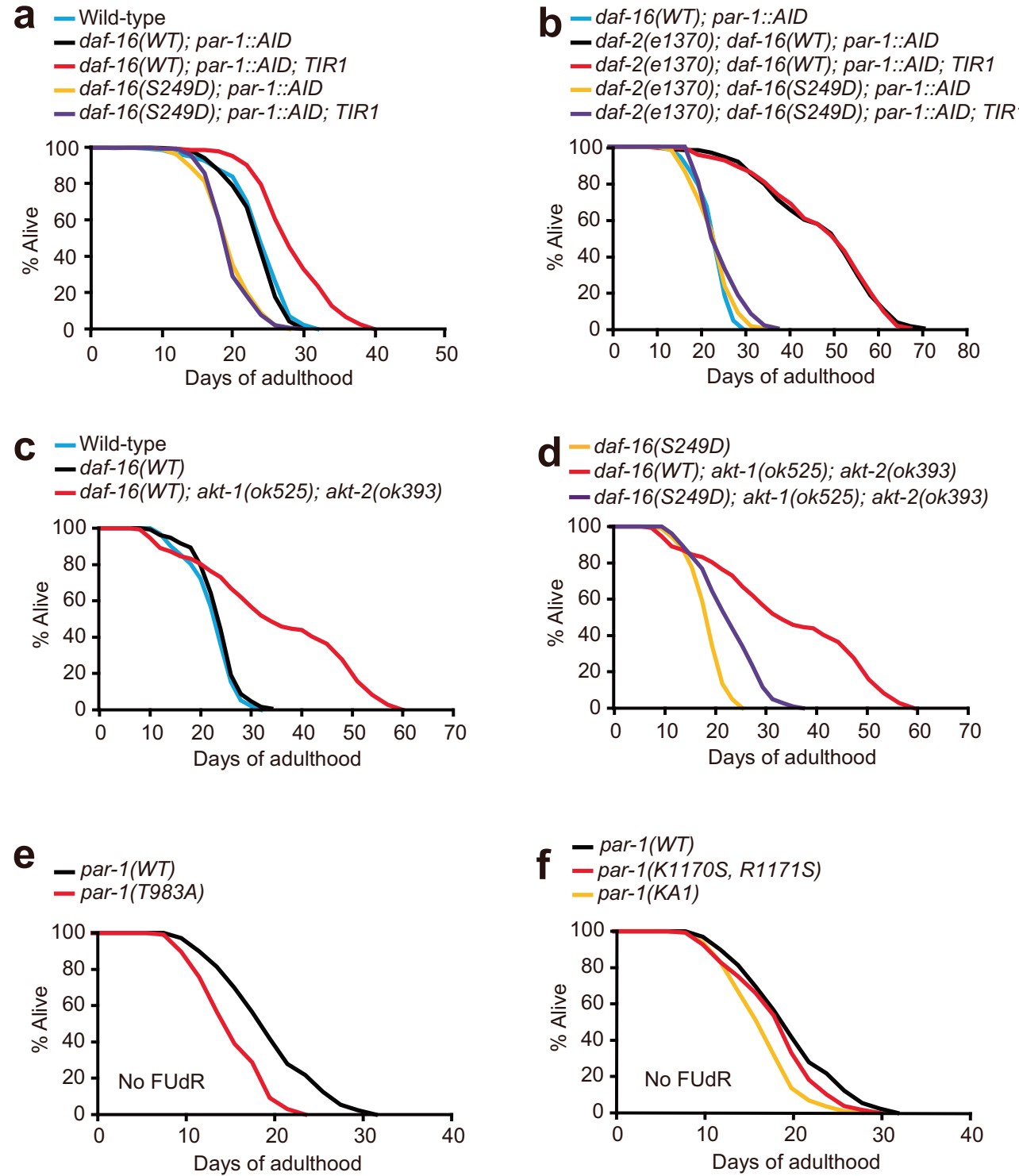

**Fig. 5 | PAR-1 participates in IIS and regulates lifespan via DAF-16. a** Lifespan assay of animals expressing *daf-16(WT)::mNG* or *daf-16(S249D)::mNG* with or without PAR-1 depletion using AID. Animals were transferred to auxin plates at the young adult stage. Depleting PAR-1 increases the lifespan of *daf-16(WT)::mNG* but not *daf-16(S249D)::mNG* animals. **b** Lifespan assay of *daf-2(e1370)* mutants expressing *daf-16(WT)::mNG* or *daf-16(S249D)::mNG* with or without PAR-1 depletion using AID. Animals were transferred to auxin plates at the young adult stage. Depleting PAR-1 does not alter the lifespan of *daf-2(e1370)* mutants regardless of the *daf-16* allele. The *daf-16(WT); par-1::AID* data are the same as in (**a**). **c** Lifespan assay of gene-edited animals expressing *daf-16(WT)::mNG* in wild-type or *akt-1(ok525); akt-2(ok393)* double mutants. **d** Lifespan assay of gene-edited animals expressing *daf-*

*16(WT)::mNG* or *daf-16(S249D)::mNG* in an *akt-1(ok525); akt-2(ok393)* double mutant background. Note that data for *daf-16(WT); akt-1(ok525); akt-2(ok393)* are the same as in (**c**). **e** Lifespan assay of gene-edited animals expressing wild-type PAR-1::mEGFP or PAR-1(T983A)::mEGFP. Phosphorylation of PAR-1 by PKC-3 at T983 was previously shown to inhibit PAR-1 kinase activity. **f** Lifespan assay of gene-edited animals expressing versions of PAR-1: PAR-1(WT)::mEGFP, PAR-1(K1170S, R1171S)::mEGFP in which two conserved basic residues in the KA1 domain are mutated to serine, and PAR-1(ΔKA1)::mEGFP which lacks the KA1 domain. Note that data for *par-1(WT)* are the same as in (**e**). See Supplementary Tables 5–7 for values and statistical analyses of lifespan assays.

**a**

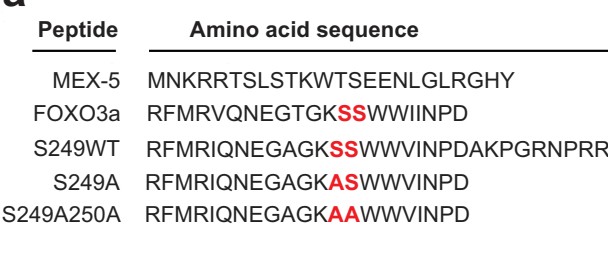

| Peptide | Amino acid sequence |
|---|---|
| MEX-5 | MNKRRTSLSTKWTSEENLGLRGHY |
| FOXO3a | RFMRVQNEGTGK**SS**WWIINPD |
| S249WT | RFMRIQNEGAGK**SS**WWVINPDAKPGRNPRR |
| S249A | RFMRIQNEGAGK**AS**WWVINPD |
| S249A250A | RFMRIQNEGAGK**AA**WWVINPD |

**b**

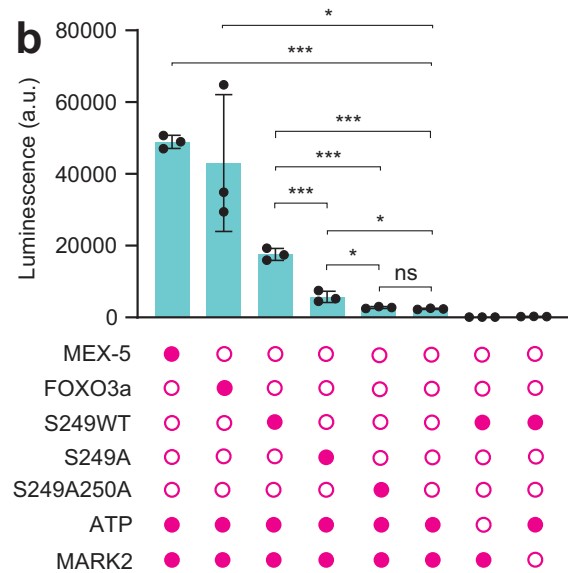

**c**

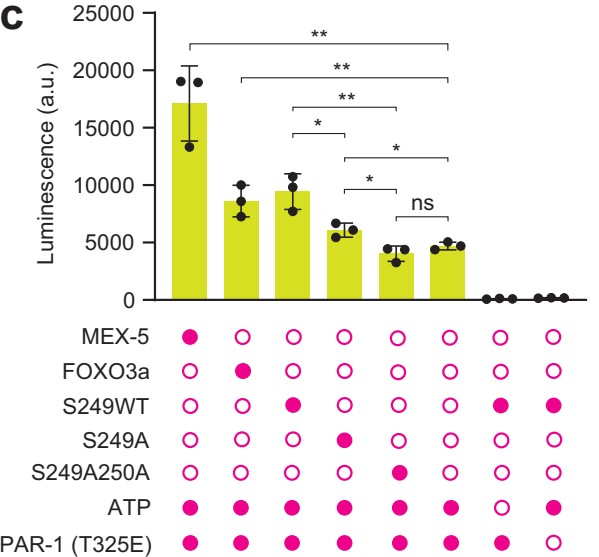

**d**

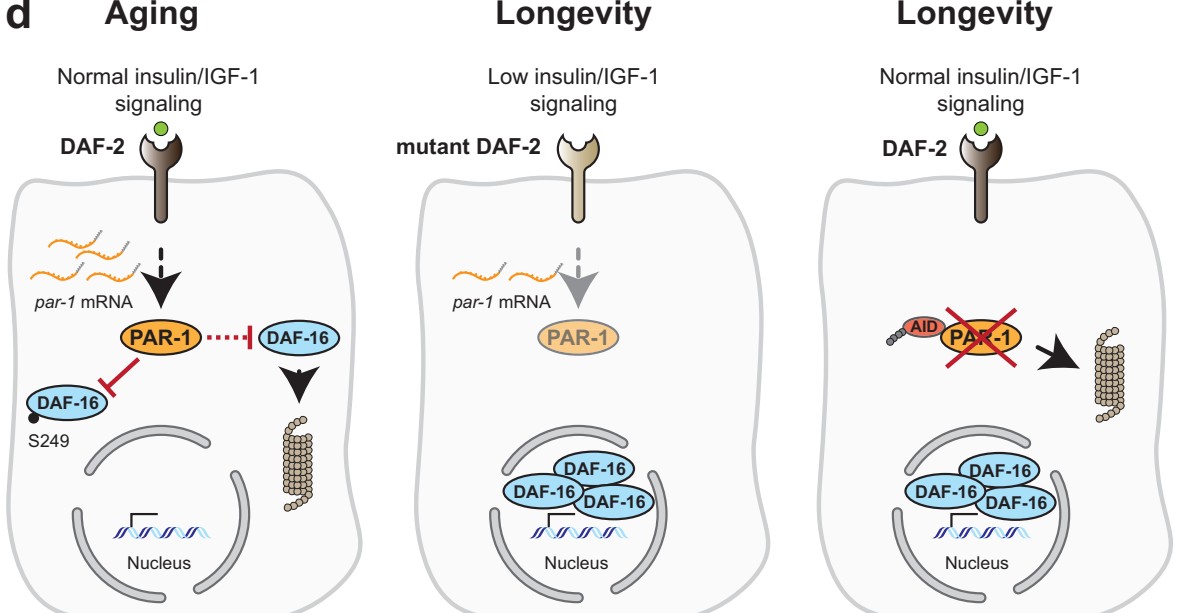

in the repair processes mediated by these proteins. Previous work in mammalian tissue culture cells with a different DNA repair protein, the ataxia-telangiectasia mutated (ATM) serine/threonine kinase, has shown that the DAF-16 ortholog FOXO3a directly binds and regulates ATM DNA repair activity[65]. Many potential interactors we identify, however, have not been studied in the context of DAF-16/FOXO function. Here, we studied the transcription factor ZFH-2 in detail,

which we find forms a complex with DAF-16 and is required for the longevity of *daf-2* mutants. The *zfh-2* locus expresses several large proteins with multiple zinc finger (zf) and homeobox (h) domains, and is the ortholog of human ZFHX3 and ZFHX4. Mutations in ZFH-2 and ZFHX3 are associated with obesity[66,67], a phenotype also associated with defects in DAF-2/insulin receptor signaling, and ZFHX3 regulates cell proliferation, differentiation and neoplastic development.

**Fig. 6 | Recombinant PAR-1 and MARK2 can phosphorylate the DAF-16 and FOXO3a S249 site in vitro. a** Amino acid sequences of peptides used for the in vitro kinase assay: MEX-5 (positive control), human FOXO3a, *C. elegans* S249WT, S249A and S249AS250A. The S249 and S250 residues (or their alanine substitutions) are highlighted. **b** In vitro kinase assay showing phosphorylation (or lack thereof) of peptide substrates by recombinant human MARK2 ($n = 3$ independent biological reactions). Error bars represent s.e.m; ns: not significant; *$p < 0.05$, ***$p < 0.001$, two-tailed Student's *t* test. **c** in vitro kinase assay showing phosphorylation (or lack thereof) of peptide substrates by recombinant activated PAR-1 (aa 1-

482, T325E) ($n = 3$ independent biological reactions). Error bars represent s.e.m; ns: not significant; *$p < 0.05$, **$p < 0.01$, two-tailed unpaired Student's *t* test. In (**b**, **c**), shaded circles indicate reagents added to the assay. **d** Model. In wild-type animals, DAF-2 signaling activates PAR-1 by an unknown mechanism. PAR-1 inhibits DAF-16 by phosphorylating it at S249 and promoting its nuclear exclusion. Separately, PAR-1 destabilizes DAF-16. Inhibition of DAF-16 de-represses *par-1* expression and restricts longevity. At low DAF-2 signaling, PAR-1 activity is low, stabilizing DAF-16 and promoting its nuclear entry, inhibiting *par-1* expression and increasing longevity. Source data are provided as a Source Data file.

Unexpectedly, disrupting *daf-2* alters expression of ZFH-2, implying a more intricate involvement of this protein in DAF-2 signaling.

In summary, we identify a signaling link between the DAF-2 insulin/IGF1 receptor, PAR-1/MARK kinases, and DAF-16/FOXO transcription factors. Since insulin/IGF signaling regulates cell metabolism, neural plasticity, and proteostasis, our observations raise the possibility that PAR-1/MARK kinases participate in these important biological processes downstream of insulin receptor signaling. Conversely, since PAR-1/MARK kinases regulate cell polarity and control microtubule dynamics, our findings suggest that DAF-2/insulin receptor signaling could regulate these processes in part by activating PAR-1.

In mammals, the four MARK kinases are expressed very broadly, and mouse mutants defective in different MARK kinases have a range of phenotypes[44]. Their functional importance is likely even broader, since different MARK kinases can, to a certain extent at least, substitute for each other. The activity of MARK kinases is highly regulated. For example, LKB1 (Liver kinase B1), TAO-1 (thousand and one) kinases, and DAPK death-associated protein kinase) activate MARK kinases[68–70] whereas GSK3b and aPKC inhibit them[47,52,71]. Our findings suggest routes by which these upstream kinases can indirectly regulate FOXO activity. Having PAR-1/MARK kinases embedded in these complex kinase networks could thus provide a brake to keep DAF-16/FOXOs in check. For example, by activating both AMPK, which activates FOXOs, and MARK kinases, which would inhibit them, LKB1 can more finely tune the activity of these important TFs.

Both MARK kinases and FOXO transcription factors are implicated in devastating human diseases, notably neurodegeneration (e.g., Alzheimer's) and cancer, and are drug targets. This makes the links we identify between these pathways potentially relevant in these disease contexts.

## Methods
### Worm strains
*C. elegans* were grown on Nematode Growth Media (NGM) plates seeded with *E. coli* OP50 as described previously[27]. For all experiments, we used animals grown in well-fed and uncrowded conditions. Animals were grown at a room temperature of 21 °C, unless otherwise stated. Some strains used in this study were obtained from the *Caenorhabditis* Genetics Center (CGC), which is funded by the NIH Office of Research Infrastructure (P40 OD010440). All *C. elegans* strains used in this study are listed in Supplementary Data 4.

### CRISPR/Cas9-mediated genome editing
Genome editing followed published CRISPR editing protocols[72]. The AX8342 strain was generated by knocking in a DNA cassette encoding TurboID::mNG::3xFLAG in frame immediately upstream of the *daf-16* stop codon. AX8623 was generated by knocking in a DNA cassette encoding mScarlet::AID::3xFLAG in frame into the 11th exon of *par-1*; exon 11 is included in all *par-1* splice isoforms (personal communication, Dr. Diane Morton). AX8625 was generated by knocking in the mScarlet::AID::3xFLAG cassette in frame immediately upstream of the stop codon of the *zfh-2* gene. AX8947 was generated by knocking in DNA encoding an mNG::HA cassette in frame immediately upstream of

the stop codon of the *daf-16* gene. AX8764 was generated by knocking in DNA encoding an mNG::3xFLAG cassette immediately upstream of the stop codon of the *mtl-1* gene. AX9134 and AX9136 were generated by mutating the codon encoding serine 249 of AX8947 *daf-16::mNG::HA* animals to one encoding alanine or aspartate, respectively. See Supplementary Data 5 for a complete list of crRNA and primer sequences used in this study.

### RNAi screen of putative DAF-16 interactors
We used RNAi to examine whether knockdown of potential DAF-16 interactors identified by TurboID altered MTL-1::mNG expression. We grew the corresponding RNAi bacterial clones from the Ahringer library[73] in 2xTY medium supplemented with Ampicillin and first confirmed plasmid identity by sequencing. We then seeded NGM plates with an overnight bacterial culture of each clone, kept the plates at room temperature for 3-4 days, then added isopropyl β-D-1-thiogalactopyranoside (IPTG) to a final concentration of 1 mM and kept them overnight at room temperature. Gravid adults from the *mtl-1::mNG::3xFLAG* or *daf-2(e1370); mtl-1::mNG::3xFLAG* gene-edited worm strains were bleached[74], the eggs transferred to the RNAi plates, and the worms harvested at the L4/YA stage. Collected worms were snap-frozen using liquid N$_2$ in 1 x NUPAGE LDS sample buffer (ThermoFisher Scientific) containing 1 x NuPAGE Sample Reducing Agent (Thermo-Fisher Scientific) and stored at − 20 °C for western blot analyses.

### Lifespan assays
Lifespan assays were conducted at 20 °C on *E. coli* OP50-seeded NGM plates. For AID knockdown experiments, both experimental and control plates contained 1 mM auxin (IAA, indole-3-acetic acid, Sigma) starting at day 1 of adulthood. Unless stated otherwise, age-synchronized pre-fertile young adult worms were transferred to plates containing 5-fluoro-2'-deoxyuridine (FUdR, Sigma-Aldrich, 5 μM) to prevent progeny growth. All lifespan assays were conducted at least twice per condition. At least four plates were used for each condition; contaminated plates were discarded. Animals that ruptured, bagged, or crawled off the plate were censored but included in the lifespan analysis as censored worms. OASIS 2 (online application for survival analysis, http://sbi.postech.ac.kr/oasis and http://sbi.postech.ac.kr/oasis2) was used for statistical analysis of the data[75], and *p*-values were calculated using the log-rank (Mantel-Cox) test. The lifespan assays and associated statistical values used in this study are listed in Supplementary Tables 1–7.

### Auxin-inducible degradation (AID)
AID assays were conducted as described[40]. Age-synchronized animals were grown on *E. coli* OP50-seeded NGM plates in the presence of 1 mM auxin (IAA, indole-3-acetic acid, Sigma). For all AID experiments involving *par-1* and *zfh-2*, including Western blot analyses, qRT-PCR analyses, mass spectrometry analyses, and microscopy of DAF-16 subcellular localization, we transferred animals to auxin plates to initiate knockdown at the L3 stage. For all AID lifespan assays involving *par-1* and *zfh-2*, we transferred the animals to auxin plates as young adults. The TIR1 version we used is from ref. 76.

## Immunoprecipitation from *C. elegans*

Co-immunoprecipitation experiments, including worm lysis, affinity purification and Western blotting, were performed as described previously[77,78], with some modifications. Briefly, synchronized populations of *C. elegans* grown on *E. coli* OP50 were harvested at the L4 or young adult stage, washed three times in M9 buffer and once with co-IP lysis buffer (50 mM HEPES, pH 7.4, 100 mM KCl, 0.5% NP40, 1 mM DTT, 1 mM PMSF and 1 complete EDTA-free proteinase inhibitor cocktail tablet per 12 ml) and frozen in co-IP lysis buffer by adding dropwise to liquid $N_2$ to obtain frozen worm 'popcorn'. A SPEX 6875D cryogenic mill was used to grind frozen *C. elegans* to a fine powder, which was then stored at $-80\,°C$. Frozen worm powder was thawed at room temperature, centrifuged for 30 minutes at 20000 x $g$ at 4 °C, and the supernatant collected. For immunoprecipitation, the lysate was incubated with anti-FLAG M2 magnetic beads (M8823 Sigma) supplemented with 10 U/ml benzonase for 1 h at 4 °C, washed three times with wash buffer (50 mM HEPES and 100 mM KCl) and eluted by heating at 90 °C in 1 x NUPAGE LDS sample buffer (ThermoFisher Scientific) containing 1 x NuPAGE Sample Reducing Agent (Thermo-Fisher Scientific) for 10 min.

## Post-translational modifications of DAF-16

Animals from indicated genetic backgrounds expressing DAF-16::mNG::HA were harvested, ground, lysed, and DAF-16::mNG::HA affinity purified for phosphorylation analyses as described previously[77,78]. Briefly, synchronized populations of *C. elegans* grown on *E. coli* OP50 were washed off NGM plates at late L3 stage using M9 buffer, collected in Falcon tubes, and transferred to NGM plates seeded with OP50 and supplemented, if necessary, with 1 mM auxin. After an 8-hour incubation, the worms were collected with M9 buffer, washed three more times in M9 buffer, once in cold 1 X TBS buffer and frozen in 1 X TBS buffer supplemented with 1 mM PMSF, 1 cOmplete EDTA-free proteinase inhibitor cocktail tablet (Roche) and 1 phosSTOP phosphatase inhibitor cocktail (Roche) per 12 ml by adding dropwise into liquid $N_2$ to obtain frozen worm 'popcorn'. A SPEX 6875D cryogenic mill was used to grind frozen *C. elegans* to a fine powder, which was then stored at $-80\,°C$. Frozen worm powder was thawed at 4 °C, placed on ice, topped with 10% NP-40 to adjust the final concentration of NP-40 to 0.5%, and the samples sonicated for 1 min using a probe sonicator microtip (QSonica 700, microtip 4417, 1.6 mm and an amplitude setting of 50/max). The samples were centrifuged at 125000 x $g$ for 30 minutes at 4 °C using a benchtop ultracentrifuge Optima MP and MLA-80 rotor (Beckman), and the clear supernatant between the pellet and surface lipid layer transferred to a new tube. The protein concentration was measured using Pierce™ 660 nm protein assay reagent supplemented with Ionic Detergent Compatibility Reagent (IDCR) (ThermoFisher Scientific). Pierce anti-HA magnetic beads (ThermoFisher Scientific) were washed three times with 1X TBS-T buffer, mixed with lysate (a total of 10 mg protein) in a 15 ml Falcon tube and incubated on a tube roller for 1 h at 4 °C. To collect beads from the lysate, a Neodymium magnet was taped to the side of the Falcon tube and incubated on a rocking platform for 15 min. The supernatant was removed, and the beads transferred to a 1.5 ml protein LoBind tube by washing with cold TBS-T. After washing the beads twice more with cold TBS-T, the buffer was removed and the beads were eluted by heating at 90 °C for 10 min in 1 x NUPAGE LDS sample buffer (ThermoFisher Scientific) containing 1 x NuPAGE Sample Reducing Agent (ThermoFisher Scientific). The entirety of the eluted samples were loaded and separated with electrophoresis using Bolt 4–12% Bis-Tris Plus gels (Thermofisher Scientific), and stained with InstantBlue (Expedeon) for visualization. Gel slices corresponding to the most prominent DAF-16 isoforms were cut out and sent for mass spectrometry analysis. 45% of the proteins obtained from every gel slice was injected into the mass spectrometer.

## Immunoblotting

Synchronized populations of *C. elegans* grown on *E. coli* OP50 were harvested at the L4 or young adult stage and washed three times in M9 buffer. The resulting packed worm suspension was adjusted to 1 x final concentration of 4 x NUPAGE LDS sample buffer (ThermoFisher Scientific) containing 1 x NuPAGE Sample Reducing Agent (ThermoFisher Scientific), and flash-frozen. The samples were then thawed, heated for 10 minutes at 90 °C, vortexed mildly for 10 minutes, centrifuged for 30 minutes at 20000 x $g$ at 22 °C and the supernatant collected. Proteins were transferred to a PVDF membrane (Thermofisher Scientific) following electrophoresis using Bolt 4–12% Bis-Tris Plus gels (Thermofisher Scientific). Membranes were blocked for 1 h at room temperature with 1% Casein blocking buffer, and incubated for 1 h at room temperature with HRP-conjugated antibodies. Membranes were then washed 3 times with TBS-T. The following antibodies were used for this study: Anti-FLAG M2-HRP mouse (1:5000 in 1% Casein buffer) (A8592 Sigma), anti-alpha tubulin-HRP mouse (1:10000 in 1% Casein buffer) (DM1A Abcam ab40742), anti-HA-HRP rat (1:5000 in 1% Casein buffer) (Clone 3F-10 Roche). Membranes were imaged using the ChemiDoc Imaging System (Model MP, Bio-Rad). Relative protein band intensities were quantified using ImageJ software (Rasband, W. S., ImageJ, U. S. National Institutes of Health, Bethesda, Maryland, USA, http://rsbweb.nih.gov/ij/).

## Light microscopy

Confocal microscopy images of gene-edited *C. elegans* expressing fluorescent proteins were acquired using a Leica (Wetzlar, Germany) SP8 inverted laser scanning confocal microscope with a 20 × 0.75 Dry objective or a 40 × 1.1 water objective, using the LAS X software platform (Leica), or using a Zeiss (Germany) LSM800 inverted laser scanning confocal microscope with a 40 × 1.2 water objective, using the Zen 3.8 software platform (Zeiss). The Z-project function in Image J (Rasband, W. S., ImageJ, U. S. National Institutes of Health, Bethesda, Maryland, USA, http://rsbweb.nih.gov/ij/) was used to obtain images. Animals were mounted on 2% agarose pads and immobilized with 12.5 μM of Levamisole.

## Subcellular localization of DAF-16::mNG

DAF-16::mNG subcellular localization was quantified as previously described[79]. Scorers were blind to genotype, and subcellular localization is shown as a percentage of total worms. Data were analyzed using GraphPad Prism version 10.0.0 for Windows (GraphPad Software, Boston, Massachusetts, USA, www.graphpad.com). *p*-values were calculated using the chi-squared test.

## Dauer assay

Dauer formation assays were performed as described previously[48,79]. Gravid adult worms were allowed to lay eggs on NGM plates at 25 °C for 5 h, to synchronize progeny. The F1 progeny were analyzed for dauer formation after 3 days of incubation at 25 °C. Dauers were scored visually using a Leica S9i dissecting stereo microscope (Leica, Wetzlar, Germany).

## Quantitative real time PCR

Total RNA was extracted from synchronized worms using Trizol (15596029, Life Tech OCI). RNA concentration was measured using a Nanodrop UV spectrophotometer (Implen) and 1-2 μl SuperScript™ IV Reverse Transcriptase (ThermoFisher Scientific). PowerTrack™ SYBR Green Master Mix (ThermoFisher Scientific) was used to perform qPCR on a Lightcycler machine (Roche 480). Tubulin was used as a reference gene for the quantification of gene expression. Primers used for qPCR experiments are listed in Supplementary Data 5.

## Brood size

Brood size was measured as described previously[79]. An L4 hermaphrodite was placed on an NGM plate seeded with OP50 and maintained

at 20 °C. Each worm was transferred to a fresh plate daily until it stopped laying eggs. The number of progeny from a single hermaphrodite that reached adulthood was counted as its brood size. Statistical analyses used Welch's unequal variances *t* test.

## Purification of recombinant PAR-1 and MEX-5

Recombinant MBP-1::PAR-1(aa 1–482, T325E) and 6xHis::MBP::MEX-5(aa 445–468) were expressed in *E. coli* Rosetta 2 cells as previously reported with some modifications[50,55]. Briefly, *E. coli* Rosetta 2 cells transformed with plasmids encoding MBP-1::PAR-1(aa 1–482, T325E) or 6xHis::MBP::MEX-5(aa 445–468) were grown at 37 °C until the $OD_{600}$ reached 0.6, then induced with 250 µM IPTG and grown overnight at 16 °C. The cells were spun at 4000 RPM at 4 °C, and the resulting cell pellet flash-frozen in liquid $N_2$ and stored at − 70 °C.

The cell pellet containing MBP-1::PAR-1(aa 1-482, T325E) was resuspended in 50 ml lysis buffer (20 mM Tris, 500 mM NaCl, 1 mM EDTA, 10% glycerol, 1 mM DTT, pH 7.4) supplemented with protease inhibitor cocktail (1:50), 1 mM $MgCl_2$, benzonase, 2 mM benzamidine and 1 mM PMSF, and incubated on ice for 30 minutes. The lysate was sonicated on ice at 40% amplitude (no microtip) for 4 min (pulse on 1 second, pulse off 4 s), centrifuged at 20.000 RPM for 30 min at 4 °C, and the supernatant collected.

Affinity chromatography of MBP-1::PAR-1(aa 1-482, T325E) used an MBPTrap affinity column (5 ml bed volume) with an Äkta Purifier 10 FPLC system. The column was equilibrated with equilibration buffer (20 mM Tris, 500 mM NaCl, 1 mM EDTA, 10% glycerol, 1 mM DTT, pH 7.4; flow rate 3 ml/min.). Supernatant containing MBP-1::PAR-1(aa 1-482, T325E) was applied to the MBPTrap affinity column (flow rate 1 ml/min.), washed with washing buffer (20 mM Tris, 500 mM NaCl, 1 mM EDTA, 10% glycerol, 1 mM DTT, pH 7.4) and eluted with elution buffer (20 mM Tris, 500 mM NaCl, 1 mM EDTA, 10% glycerol, 1 mM DTT,10 mM maltose pH 7.4). Pooled fractions were loaded onto a HiLoad 16/600 Superdex column and run using Äkta 9 (flow rate 0.5 ml/min.). Collected samples were frozen at − 70 °C.

The cell pellet containing 6xHis::MBP::MEX-5(aa 445–468) was resuspended in 50 ml lysis buffer (20 mM HEPES, 200 mM NaCl, 20 mM imidazole, 1 mM TCEP, pH 8) supplemented with protease inhibitor cocktail (1:50), 1 mM $MgCl_2$, benzonase, 2 mM benzamidine, and incubated on ice for 30 min. The lysate was sonicated on ice at 40% amplitude (no microtip) for 4 minutes (pulse on 1 s, pulse off 4 s), centrifuged at 20.000 RPM for 30 min at 4 °C and the supernatant collected.

Affinity chromatography of 6xHis::MBP::MEX-5(aa 445–468) used a HisTrap affinity column (5 ml bed volume) with an Äkta Purifier 10 FPLC system. The column was equilibrated with equilibration buffer (20 mM HEPES, 200 mM NaCl, 20 mM imidazole, 1 mM TCEP, pH 8; flow rate 3 ml/min.). Supernatant containing 6xHis::MBP::MEX-5(aa 445–468) was applied to a HisTrap affinity column (flow rate 2 ml/min.), washed with washing buffer (20 mM HEPES, 500 mM NaCl, 20 mM imidazole, 0.2% w/v Triton-X100, 1 mM TCEP, pH 8) and eluted with elution buffer (20 mM HEPES, 200 mM NaCl, 250 mM imidazole, 1 mM TCEP, pH 7.6). Pooled fractions were loaded onto a Superdex 200 Increase 10/300 column and run with an Äkta 10 (flow rate 0.5 ml/min.). Collected samples were frozen at − 70 °C.

## In vitro kinase assay

The MARK2 kinase enzyme activity detection kit, utilizing the ADP-Glo™ system (VA7215, Promega), was used for in vitro kinase assays which were performed according to the manufacturer's protocol. Briefly, recombinant human MARK2 or *C. elegans* PAR-1 was diluted to 30 ng/µl in 1x kinase buffer D (40 mM Tris pH 7.5, 20 mM $MgCl_2$, 0.1 mg/ml BSA, 50 µM DTT, 1% DMSO). 2.5x ATP/peptide substrate mix was prepared so as to contain 0.5 µg/µl peptide and 25 µM of ATP in 2.5x kinase buffer D. 2 µl of the 2.5x ATP/substrate mix was transferred to the wells of a 384-well white plate (200 ng/µl peptide and 10 µM of

ATP final reaction concentration). 1 µl of 1x kinase buffer D and 2 µl of MARK2 or PAR-1 in 1x kinase buffer D (60 ng of enzyme per well) were added to the wells and incubated at room temperature for 40 minutes. 5 µl of ADP-Glo™ reagent was added to the wells and the plate was incubated for another 40 minutes at room temperature. Lastly, 10 µl of Kinase Detection Reagent was added to the wells, the plate was incubated for 30 minutes at room temperature, and luminescence recorded using a BMG Clariostar plate reader.

## TurboID proximity labeling and extraction of biotinylated proteins from *C. elegans*

Gravid adult worms were bleached, and around 20,000 eggs transferred to each of 15–20 90 mm NGM plates seeded with *E. coli* MG1655, to obtain synchronized populations of well-fed unstressed worms. The animals were harvested at L4 or young adult stage, washed three times in M9 buffer, and incubated at room temperature (21 °C) in M9 buffer supplemented with 1 mM biotin and *E. coli* MG1655 for 2 h unless stated otherwise. The worms were then washed 3x in M9 buffer and allowed to settle on ice. After completely aspirating the M9 buffer, an equal volume of 2x TBS buffer supplemented with 1 mM PMSF and cOmplete EDTA-free protease inhibitor cocktail (Roche Applied Science) was added to the packed worms. The animals were again allowed to settle on ice and then added dropwise to liquid $N_2$ to obtain frozen worm 'popcorn'. A Spex 6875D cryogenic mill was used to grind the *C. elegans* popcorn to a fine powder, which was then stored at − 80 °C. A total of 12.5 mg of protein was used for affinity purification of biotinylated proteins for each sample. Protein extraction, depletion of endogenously-biotinylated protein, and affinity purification of biotinylated proteins was performed as described previously[28].

## Bioinformatic analyses and software

Gene ontology (GO) analyses used custom software available at https://uniprot-batch-requester.science.ista.ac.at/. Volcano plots were made using Amica software version 3.0.1[80]. Data in Fig. 1e, f, were plotted using Cytoscape software (https://cytoscape.org). Figures were prepared using Adobe Illustrator version 28.5. GraphPad Prism 10 was used to generate bar graphs and statistical analyses.

## Statistics and reproducibility

The sample size used in each experiment was not predetermined or formally justified for statistical power. Sample sizes used in our experiments followed the conventions used in the field. No data were excluded from analysis. The number of replicates for each experiment is indicated in the relevant figure legend or supplemental table. For all mass spectrometry, Western blot and co-IP analyses, entire populations of worms were used. For microscopy imaging, DAF-16 subcellular localization and lifespan assays, random animals at the indicated developmental stages were picked for experiments. The experiments in Figs. 3g, 4d, e were performed blind.

## On bead digestion for Mass spectrometry analysis of DAF-16 proximity labeling

Streptavidin beads were resuspended in 50 µL 1 M urea and 50 mM ammonium bicarbonate. Disulfide bonds were reduced with 2 µL of 250 mM dithiothreitol (DTT) for 30 min at room temperature before adding 2 µL of 500 mM iodoacetamide and incubating for 30 min at room temperature in the dark. The remaining iodoacetamide was quenched with 1 µL of 250 mM DTT for 10 min. Proteins were digested with 150 ng LysC (mass spectrometry grade, FUJIFILM Wako chemicals) at 25 °C overnight. The supernatant was transferred to a new tube and digested with 150 ng trypsin (Trypsin Gold, Promega) in 1.5 µL 50 mM ammonium bicarbonate at 37 °C for 5 h. The digest was stopped by the addition of trifluoroacetic acid (TFA) to a final concentration of 0.5 %, and the peptides desalted using C18 Stagetips[81].

## Liquid chromatography-Mass spectrometry data acquisition of DAF-16 proximity labeling

Peptides were separated on an Ultimate 3000 RSLC nano-flow chromatography system (ThermoFisher), using a pre-column for sample loading (Acclaim PepMap C18, 2 cm × 0.1 mm, 5 μm, ThermoFisher), and a C18 analytical column (Acclaim PepMap C18, 50 cm × 0.075 mm, 2 μm, ThermoFisher), applying a segmented linear gradient from 2% to 35% and finally 80% solvent B (80 % acetonitrile, 0.1 % formic acid; solvent A 0.1 % formic acid) at a flow rate of 230 nL/min over 120 min.

Eluting peptides were analyzed on an Exploris 480 Orbitrap mass spectrometer (ThermoFisher) coupled to the column with a FAIMS pro ion-source (ThermoFisher) using coated emitter tips (PepSep, MSWil) with the following settings: The mass spectrometer was operated in DDA mode with two FAIMS compensation voltages (CV) set to − 45 or − 60 and 1.5 s cycle time per CV. The survey scans were obtained in a mass range of 350-1500 m/z, at a resolution of 60 k at 200 m/z, and a normalized AGC target at 100%. The most intense ions were selected with an isolation width of 1.2 m/z, fragmented in the HCD cell at 28% collision energy, and the spectra recorded for max. 100 ms at a normalized AGC target of 100% and a resolution of 15k. Peptides with a charge of + 2 to + 6 were included for fragmentation, the peptide match feature was set to preferred, the exclude isotope feature was enabled, and selected precursors were dynamically excluded from repeated sampling for 45 s.

## Data analysis of DAF-16 proximity labeling

MS raw data, split for each CV using FreeStyle 1.7 (ThermoFisher), were analyzed using the MaxQuant software package (version 2.1.0.0)[82] with the Uniprot *Caenorhabditis elegans* reference proteome (version 2022_02, www.uniprot.org), as well as a database of most common contaminants. The search was performed with full trypsin specificity and a maximum of two missed cleavages at a protein and peptide spectrum match false discovery rate of 1%. Carbamidomethylation of cysteine residues was set as fixed, and oxidation of methionine and N-terminal acetylation as variable modifications. For label-free quantification, the "match between runs" only within the sample batch and the LFQ function were activated - all other parameters were left at default.

MaxQuant output tables were further processed in R 4.2.1 (https://www.R-project.org) using Cassiopeia_LFQ (https://github.com/maxperutzlabs-ms/Cassiopeia_LFQ). Reverse database identifications, contaminant proteins, protein groups identified only by a modified peptide, protein groups with less than two quantitative values in one experimental group, and protein groups with less than 2 razor peptides were removed from further analysis. Missing values were replaced by randomly drawing data points from a normal distribution model on the whole dataset (data mean shifted by − 1.8 standard deviations, a width of the distribution of 0.3 standard deviations). Statistical analysis comparing experiments was performed using the Linear Models for Microarray Analysis (LIMMA, version 3.54.2)[83] package in R. Moderated t-statistics were calculated using the limma-trend method with batch correction using the replicate number as batch, and multiple testing correction was applied using the Benjamini-Hochberg (BH) method.

## Sample preparation for mass spectrometry analysis of DAF-16 phosphorylation

Coomassie stained gel bands were cut and destained with a mixture of acetonitrile (ACN) and 50 mM ammonium bicarbonate. Disulfide bridges were reduced using dithiothreitol and free SH-groups were subsequently alkylated by iodoacetamide. The digestion with trypsin (Trypsin Gold, Mass Spec Grade, Promega, V5280) was carried out overnight at 37 °C, while the digestion with chymotrypsin was carried out at 25 °C for 5 hours. Digestion was then stopped by adding 10% formic acid to a final concentration of approximately 5%. Peptides were extracted from the gel with 5% formic acid by repeated sonication.

## Liquid chromatography-mass spectrometry analysis of DAF-16 phosphorylation

45% of the gel-extracted peptides were used for analysis. Peptides were separated on an Ultimate 3000 RSLC nano-flow chromatography system (ThermoFisher), using a pre-column for sample loading (Acclaim PepMap C18, 2 cm × 0.1 mm, 5 μm, ThermoFisher), and a C18 analytical column (Acclaim PepMap C18, 50 cm × 0.75 mm, 2 μm, ThermoFisher), applying a segmented linear gradient from 2% to 35% and finally 80% solvent B (80 % acetonitrile, 0.1 % formic acid; solvent A 0.1 % formic acid) at a flow rate of 230 nL/min over 60 min.

Eluting peptides were analyzed on an Exploris 480 Orbitrap mass spectrometer (ThermoFisher) coupled to the column using coated emitter tips (PepSep, MSWil) with the following settings. The mass spectrometer was operated in DDA mode with a cycle time of 2 s. The survey scans were obtained in a mass range of 375–1500 m/z, at a resolution of 120 k at 200 m/z, and a normalized AGC target at 300%. The most intense ions were selected with an isolation width of 1.2 m/z, fragmented in the HCD cell at 28% collision energy, and the spectra recorded for max. 200 ms at a normalized AGC target of 200% and a resolution of 15 k. Peptides with a charge of + 2 to + 6 were included for fragmentation, the peptide match feature was set to preferred, the exclude isotope feature was enabled, and selected precursors were dynamically excluded from repeated sampling for 20 s.

## Data analysis of DAF-16 phosphorylation

The RAW MS data were first analyzed with FragPipe (20.0), using MSFragger (3.8)[84], IonQuant (1.9.8)[85], and Philosopher (5.0.0)[86]. The default FragPipe workflow for label free quantification (LFQ-MBR) was used, except "Normalize intensity across runs" was turned off. Cleavage specificity was set to Trypsin/P, with two missed cleavages allowed. The protein FDR was set to 1%. A mass of 57.02146 (carbamidomethyl) was used as fixed cysteine modification; methionine oxidation and protein N-terminal acetylation were specified as variable modifications. MS2 spectra were searched against the *Caenorhabditis elegans* 1 protein per gene reference proteome from Uniprot (Proteome ID: UP000001940, release 2023.03), concatenated with a database of 382 common laboratory contaminants (release 2023.03, https://github.com/maxperutzlabs-ms/perutz-ms-contaminants).

For the post-translational modification (PTM) analysis, the raw MS data was searched against 495 proteins identified in the 1st search with more than 10 combined spectra count in FragPipe (20.0), using the same settings, with additional variable modifications including phosphorylation (STY), GlyGly(K), acetylation(K) and methylation(K). MS/MS spectra of a selection of identified modified peptides were manually validated.

Computational analysis was performed using Python and the in-house developed Python library MsReport (version 0.0.23). Only non-contaminant proteins identified with a minimum of two peptides and quantified in at least three replicates of one experiment were considered for the analysis. LFQ protein intensities reported by FragPipe were log2-transformed and normalized to the DAF-16 protein intensity using the MedianNormalizer from MsReport. Missing values were imputed by drawing random values from a normal distribution. Sigma and mu of this distribution were calculated per sample from the standard deviation and median of the observed log2 protein intensities ($\mu$ = median sample LFQ intensity − 1.8 standard deviations of the sample LFQ intensities, $\sigma$ = 0.3 × standard deviation of the sample LFQ intensities).

Statistical analysis comparing experiments was performed using the Linear Models for Microarray Analysis (LIMMA, version 3.54.2)[83] package in R. Moderated t-statistics were calculated using the limma-trend method with batch correction using the replicate number as batch, and multiple testing correction was applied using the Benjamini-Hochberg (BH) method. The in-house Python library XlsxReport (0.1.0) was used to create a formatted Excel file summarizing the results of protein quantification (Supplementary Data 3).

**Ethics statement**

This work used the free-living nematode *C. elegans*, for which there is no requirement for review and approval from an institutional animal care and use committee. Gene editing and transgenic experiments were carried out following ISTA guidelines for such work.

**Reporting summary**

Further information on research design is available in the Nature Portfolio Reporting Summary linked to this article.

## Data availability

Source data are provided with this paper. The mass spectrometry proteomics data for DAF-16 phosphorylation analysis have been deposited at the ProteomeXchange Consortium via the PRIDE[87] partner repository with the data set identifier PXD053590. The mass spectrometry proteomics data for DAF-16 proximity labeling analysis have been deposited at the ProteomeXchange Consortium via the PRIDE[87] partner repository with the data set identifier PXD053591. Source data are provided in this paper.

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

## Acknowledgements

We thank de Bono lab members for helpful comments on the manuscript, and the Mass Spec Facility at the Max Perutz Labs, notably Wei-Qiang Chen and Markus Hartl, for invaluable discussions and comments on mass spec analyses of worm samples. All LC-MS/MS analyses were performed on instruments of the Vienna BioCenter Core Facilities (VBCF). Microscopy was supported by the Scientific Services Units (SSU) of ISTA through resources provided by the Imaging & Optics Facility (IOF). We are grateful to Dr. Geraldine Seydoux (Johns Hopkins University) for worm strains and plasmids, and Dr. Seung-Jae V. Lee (KAIST) for RNAi clones. We are grateful to Ekaterina Lashmanova for designing the *daf-16::TbID::mNG::3xFLAG* knock-in construct and for her outstanding support in the lab. This work was supported by a Wellcome Investigator Award (209504/A/17/Z) to MdB and an ISTplus Fellowship to MA (Marie Sklodowska-Curie agreement No 754411).

## Author contributions

M.A. and M.d.B. conceived the project and designed the experiments; M.A. and H.S. performed experiments; M.A., H.S., and M.d.B. analyzed data; M.A. and M.d.B. wrote the manuscript.

## Competing interests

The authors declare no competing interests
