## [Transparent Peer Review file · Nature Communications]

Proximity labeling of DAF-16 FOXO highlights aging regulatory proteins

Corresponding Author: Professor Mario de Bono

Version 0:

Reviewer comments:

Reviewer #1

(Remarks to the Author)

“Proximity labeling of DAF-16 FOXO highlights aging regulatory proteins” is a technically sound, well-documented study. The data are of high quality. The most informative finding is about PAR-1; specifically, PAR-1 the kinase inhibits DAF-16 by phosphorylating DAF-16 at residue S294. Loss of *daf-2* function can extend lifespan only modestly if DAF-16(WT) is replaced with the DAF-16(S294D) mutant version, which mimics phosphorylated DAF-16 by PAR-1. However, the manuscript does not read like a coherent study because it drifts away before and after the PAR-1 investigation. Additionally, the regulation of DAF-16 by PAR-1 is not connected to that by insulin signaling—are they independent of each other? Nutrient availability regulates DAF-16 through insulin signaling. Under what physiological conditions is DAF-16 regulated by PAR-1? For the above reasons, although the authors have done a lot, the manuscript still feels incomplete and lacks a clear take-home message.

Major Concerns:

1. Potential interactors of DAF-16 found by proximity labeling should be compared with the known interactors of DAF-16 (e.g., FTT-2, AKT, MAT-33, SIR-2.1, SWI/SNF, TAX-6, UNC-43...) as well as other potential interactors reported in earlier studies using different omics methods. The advantages and biases of proximity labeling and other methods that have been used to identify DAF-16 interacting proteins need be discussed. The manuscript, in its current version, gives the impression that no such studies have been conducted.
2. Among the potential interactors of DAF-16 found by proximity labeling, which ones are downstream effectors of insulin signaling, for instance, transcriptional targets of DAF-16? Are they up- or down-regulated in the *daf-2* mutant?
3. The investigation of PAR-1 is more in-depth than that of ZFH-2, which in turn is more in-depth than that of MTCL-1, a presumed microtubule-crosslinking factor. I suggest that the MTCL-1 section be removed. Does the degradation of MTCL-1 lead to impairment of microtubules? Is this verified? Additionally, there is no verification data showing physical interaction between MTCL-1 and DAF-16, or that the regulation of DAF-16 by MTCL-1 is direct. DAF-16 is extremely sensitive to stress of all kinds; it translocates readily to the nucleus when a worm is removed from its most comfortable conditions. Nuclear accumulation of DAF-16 induced by MTCL-1 AID could be interpreted as DAF-16 responding to stress caused by impairment of the microtubule network.
4. An in vitro kinase assay is needed to test whether PAR-1 can directly phosphorylate DAF-16 at S249.

Minor Issues:

1. “Interactors” identified by proximity labeling should be “potential interactors.”
2. The authors show that ZFH-2 negatively regulates *mtl-1*. Does ZFH-2 regulate *mtl-1* expression directly or through DAF-16? How can we reconcile the lifespan phenotype caused by ZFH-2 degradation?
3. *zfh-2*, *par-1*, and *mtcl-1* all encode multiple isoforms. Schematic diagrams showing the knock-in sites of mSc::AID::3*FLAG tag would be more informative than textual descriptions.
4. The gene name of *mtcl-1* in Wormbase is tag-241.

Reviewer #2

(Remarks to the Author)

Artan et al present an interesting application of the turboID proximity labeling system, which they use to identify direct protein interactors of DAF-16. *daf-16* is an evolutionary conserved transcription factor of great general interest, and a protein that has been intensively studied in the context of *C. elegans* aging for over thirty years. Previous systematic studies of *daf-16*

interactors have focused primarily on its direct and indirect transcriptional targets (e.g Murphy et al 2003 10.1038/nature01789), and a small number of daf-16 regulators identified using genetic techniques. A systematic method for identifying direct DAF-16 interactors was previously developed, involving IP pull-down of interactors using DAF-16 as bait, followed by mass spectrometry (Riedel 2013 10.1038/ncb2720). turboID has several potential advantages over this previous IP pulldown method, including increased sensitivity and the ability to better capture *in vivo* interactions that may not persist *ex vivo*. These potential advantages provide good motivation for the application of turboID to study DAF-16, as done here by Artan et al.

The authors succeed in identifying several novel DAF-16 interactors. In particular, they quite extensively validate PAR-1 as a physiologic regulator of DAF-16, going so far as to identify the specific position on DAF-16 that is modified during PAR-1's post-translational regulation of DAF-16. The authors also identify several other proteins that appear to regulate DAF-16 localization. The authors confirm the physiologic relevance of these interactors by demonstrating that their knockdown modulates DAF-16's influence on lifespan.

The authors present their findings clearly using sound arguments. Where the current manuscript falls bit short is in clarifying exactly how the field benefits from the discovery of these additional mechanistic interactors of DAF-16. Knockdown of DAF-16 has, for twenty years, been known to differentially regulate double-digit percentages of all genes expression (Murphy et al 2003), most of which remain poorly understood. The post-translational regulation of DAF-16 is already known to be rather complex, involving acetylation (Ao-Lin Hsu et al PLOS Genetics 2012 10.1371/journal.pgen.1002948), ubiquitination (Heimbucher et al Cell Metabolism 2015 10.1016/j.cmet.2015.06.002) and a variety of other interactors (Yen, Antioxid Redox Sig 2011, 10.1089/ars.2010.3490). How does the addition of four or five additional regulatory mechanisms to the existing set further our understanding of DAF-16 and its physiologic role? How are the newly discovered protein interactors unique, different, or important in the context of the large body of existing knowledge of DAF-16 function? Clarification of these questions would also help demonstrate that turboID is a technology that can provide novel biological insights.

Major points

1. The mechanistic characterization of DAF-16 interactions appears solid (with a few small caveats below). The lifespan experiments establish the physiologic relevance of the physical interactions described, which are mediated by the well-known influence of DAF-16 localization on organismal gene-expression and aging. However, the authors are mostly silent on what the new regulatory factors teach us beyond the simple fact of their existence. The manuscript would be more compelling to a broader audience were it clearer what generalizable knowledge the paper contributes.
2. The authors find that DAF-16 protein abundance is upregulated 2-fold by PAR-1 knockdown (Fig. 3 h-i). The authors should clarify whether they believe this is a regulatory mechanism separate from PAR-1's activity to promote phosphorylation at DAF-16 residue S249, or not. They state that "not all effects of PAR-1 knockdown are mediated via phosphorylation of S249" but then show data that the change in protein abundance seems to require the wild-type s249 residue. If the influence of PAR-1 on DAF-16 abundance requires an intact phosphorylation site, then the change in DAF-16 abundance affected by PAR-1 would then seem mediated via phosphorylation, no? What do the authors think is going on here?
3. In some places, the author's logic in providing a mechanistic rationale for identifying specific proteins as DAF-16 interactions seem a bit weak. For example, in the case of APT-9 the authors state that "These proteins are implicated in vesicular traffic and microtubule function, hinting at their possible involvement in keeping DAF-16 outside the nucleus when insulin signaling is high." However, it is not clear to this reviewer how APT-9's role in vesicular trafficking would suggest that it should keep DAF-16 outside the nucleus. Do the authors believe that all proteins involved in vesicular trafficking should be involved in DAF-16 regulation?
4. The logic for selecting specific hits from the turboID analysis for follow-up experimental is currently unclear. In retrospect, the choice of PAR-1 and other interactors appears well-justified, but the authors could not have known this would be the case in advance. What criteria were used to prioritize PAR-1 and the other interactors studied, over other hits with similar statistical support in the turboID data? A clearer rationale would provide important context for the Par-1 results, while also helping future users of turboID will be in a similar position trying to select promising targets from a list of turboID interactors.

For example: in Fig. 1d there are many statistically significant hits with large effect sizes that are unlabeled and remain unmentioned in the manuscript. Looking at the supplementary tables, many potentially interesting interactors are present, for example genes involved in transcription (eg F57B10.8, taf-3) or nuclear import (eg MEL28). Is there a reason these targets were not followed up upon? For example, maybe these hits would appear for any transcription factor characterized via turboID, and so are not interesting DAF-16-specific interactors?. Other hits from the turboID experiment that remain unmentioned include SYD-1 (protein Q86NH1), which seems to be an interesting gene involved in neuronal development and G5ECF6, a transposon-derived protein. Are these DAF-16 interactors worth following up on?
5. On a related note, the authors highlight nuclear proteins whose interactions increase in daf-2 mutants as well as cytoplasmic proteins whose interactions decrease in daf-2 mutants. Such proteins fit the narrative of DAF-2 activating DAF-16 primarily by promoting its nuclear localization. However, is it the case that not a single protein showed an opposite-than-expected effect—e.g, a cytoplasmic protein that increased in biotinylation after DAF-2 knockdown, or a nuclear protein that decreased in biotinylation after DAF-2 knockdown? Such proteins might be very interesting.
6. One more area that seems to be relatively unexplored in the manuscript is the potential for DAF-16 to have functional

interactions with cytoplasmic proteins that are not regulators of DAF-16. The authors highlight several cases where cytoplasmic proteins modify DAF-16 and serve to regulate DAF-16, but is this the exclusive role of DAF-16 interactions in the cytoplasm? Could the causal relationship be reversed for any of the new identified cytoplasmic biotinylation targets— could they be regulated by DAF-16, and not vice-versa?

Reviewer #3

(Remarks to the Author)

Artan and coauthors reported the use of proximity labeling of DAF-16/FOXO in *C. elegans*. This study is interesting and provides new insights into the function of DAF-16/FOXO and the use of cytoplasmic and nuclear conditions to identify DAF-16 interactors enriched in different subcellular locations and their functions.

The reviewer was asked by the editor to evaluate the quality of the proteomic experiments, but many details are missing in the Methods section, making it difficult for the reviewer to properly assess the quality of the experiments.

1. The authors cited their previously published manuscripts (refs 15 and 16) as references for the proximity labeling method; however, the reviewer still finds certain aspects unclear. For example, why was the WT control used instead of the TurboID control described in the references? What developmental stage of the worms was used for each strain, and why? How many plates or worms were used? What was the total amount of protein used for enrichment? What imputation method was used?
2. Similarly, additional details are needed on how DAF-16 phosphorylation was determined. What was the amount of input lysate? What was the amount of purified DAF-16? How much was loaded onto the gel? Please also provide the MS2 spectra for the selected phosphorylation sites.

Reviewer #4

(Remarks to the Author)

Version 1:

Reviewer comments:

Reviewer #1

(Remarks to the Author)

The authors have addressed all of my concerns with new experiments, well presented data, and careful adjustment of the text. The revised manuscript is nicely organized and put into the context of related studies in the literature. I would like to congratulate the authors for this nice study.

Reviewer #2

(Remarks to the Author)

The authors have satisfied all of my concerns except for one related to major point #4. The authors state pragmatic reasons for selecting only a few specific hits from their biochemical assay for validation and mechanistic follow-up. These reasons do seem justifiable. However, future reader's interests may differ from those of the authors, and right now the manuscript seems to make it unnecessarily hard to ascertain the identity of the significant hits not chosen for follow-up. This is in part because Fig 1 d omits labels for most genes,, including several seemingly excellent candidates. For example, what is the name of the significant hit below TMYB-1 and to the upper- left of PAR-1? The two genes to the left of SWSN-3 ?

This problem would seem to be easily solved by including a discrete table that lists all the data plotted in Fig. 1 d, with genes sorted by effect size and significance, ideally as a sortable excel table or CSV file so interested readers can easily poke through. Making the complete gene list + statistics more accessible would also dispel any impression that the authors might be hiding something here.

Reviewer #3

(Remarks to the Author)

The authors have addressed my questions fully. Great job!

Reviewer #4

(Remarks to the Author)

REVIEWER COMMENTS

Reviewer #1 (Remarks to the Author):

“Proximity labeling of DAF-16 FOXO highlights aging regulatory proteins” is a technically sound, well-documented study. The data are of high quality.

We thank our referee for their supportive feedback.

The most informative finding is about PAR-1; specifically, PAR-1 the kinase inhibits DAF-16 by phosphorylating DAF-16 at residue S294. Loss of *daf-2* function can extend lifespan only modestly if DAF-16(WT) is replaced with the DAF-16(S294D) mutant version, which mimics phosphorylated DAF-16 by PAR-1. However, the manuscript does not read like a coherent study because it drifts away before and after the PAR-1 investigation.

We have sought to make the manuscript tighter by making multiple changes. Most significantly, we removed the entirety of the MTCL-1 section. Our reviewer is right however, that while elucidating PAR-1 regulation of DAF-16 forms the core of the work, this is not the only message. An additional goal is to provide a resource, allowing others to dip into the DAF-16 proximity labelling data and investigate leads they find interesting. Inevitably, this means we are weaving several strands. The PAR-1 and ZFH-2 stories provide proof-of-principle that proximity labelling identifies previously unknown interactors relevant to DAF-16 biology. However, both in the main figures and in the supplementary data we present additional potential DAF-16 interactors, and although we do not follow these up, we need to mention them. We have tried to explicitly highlight that these data provide a resource while being brief.

Additionally, the regulation of DAF-16 by PAR-1 is not connected to that by insulin signaling—are they independent of each other?

The genetic and biochemical data suggest that DAF-2 insulin receptor signalling activates PAR-1 (most likely indirectly), and that PAR-1 then inhibits DAF-16 directly by phosphorylation: DAF-2 - PAR-1 --| DAF-16

The genetic evidence supporting this claim is that PAR-1::AID knockdown does not extend the lifespan of *daf-2* mutants (Fig. 3k), and neither does a *daf-16(S249A)* allele (Fig 4h). This non-additivity suggests DAF-2 and PAR-1 act in the same pathway to control longevity. Moreover, the lifespan extension conferred by knockdown or inhibition of PAR-1, like that conferred by *daf-2* mutations, depends on functional DAF-16 (Fig. 3j, l), suggesting that DAF-2 activates PAR-1 which inhibits DAF-16.

The biochemical data support this model. First, mass spectrometry data show that phosphorylation of DAF-16 at the S249 residue is significantly reduced when either DAF-2 or PAR-1 is defective (Fig. 4a). In other words, both DAF-2 and PAR-1 promote

phosphorylation at S249, consistent with the genetic data, and with a model in which DAF-2 activates PAR-1. Further support comes from a recent study (Li et al, *Nature Communications* 2021 12:1-16) that examined how the phosphoproteome of well-fed unstressed *C. elegans* (young adults) changes in *daf-2* mutants. They reported reduced phosphorylation of predicted PAR-1 substrates (see Fig. 6a in the Li et al paper), consistent with generally reduced PAR-1 kinase activity in *daf-2* mutants. We refer to this work in the discussion.

The TurboID data suggest PAR-1 and DAF-16 are in close proximity *in vivo* (Fig. 1d), and DAF-16 and PAR-1 co-immunoprecipitate from *ex vivo* *C. elegans* extracts (Fig. 3e), suggesting they form a complex. In our revised manuscript we have added *in vitro* data indicating that both PAR-1 and its human ortholog, MARK2 kinase, can phosphorylate S249 (Fig. 6a-c). We also show that PAR-1 knockdown doubles DAF-16 protein levels (Fig. 3h, i; Supplementary Fig. 5e, f) by a mechanism that does not alter *daf-16* mRNA levels (Supplementary Fig 3d) and that is independent of phosphorylation at S249 (Supplementary Fig. 5e, f)

We acknowledge however, that we have not established how DAF-2 activates PAR-1. This is a question requiring further biochemistry in future work.

Nutrient availability regulates DAF-16 through insulin signaling. Under what physiological conditions is DAF-16 regulated by PAR-1?

All our experiments that are relevant for this question were performed using animals grown in well-fed and uncrowded conditions at 20 - 21 °C. This includes our proximity labelling, the PAR-1::AID knockdown, and the lifespan experiments. Our data thus suggest that PAR-1 negatively regulates DAF-16 under standard, unstressed conditions. We now state this explicitly in the manuscript.

For the above reasons, although the authors have done a lot, the manuscript still feels incomplete and lacks a clear take-home message.

We sincerely hope that the changes we have made following our reviewer's advice have improved the clarity of the manuscript.

Major Concerns:

1. Potential interactors of DAF-16 found by proximity labeling should be compared with the known interactors of DAF-16 (e.g., FTT-2, AKT, MAT-33, SIR-2.1, SWI/SNF, TAX-6, UNC-43...) as well as other potential interactors reported in earlier studies using different omics methods. The advantages and biases of proximity labeling and other methods that have been used to identify DAF-16 interacting proteins need be discussed. The manuscript, in its current version, gives the impression that no such studies have been conducted.

We sincerely apologise if in our first submission we inadvertently gave the impression that no studies have been carried out to identify DAF-16 interacting proteins. Our reviewer makes several very good suggestions here.

Prompted by their comments, we have surveyed the literature to highlight DAF-16 interactors or regulators identified in previous studies. We then created a Table that provides the proximity labelling data for each of these proteins as well as referring to the papers in which their interaction with DAF-16 was identified. We have sought to be inclusive in our analysis, but apologise in advance if we have omitted any paper.

We now say:

By searching the literature we created a list of 17 proteins that are not listed in Fig. 1e,f but are known to co-IP with DAF-16. We could detect and quantify 12 of these proteins in our proximity labelling data. Six proteins, SIR-2.1, HCF-1, SGK-1, AKT-1/2, and PRMT-1, were significantly enriched proximal to DAF-16 compared to control (Supplementary Table 2). Two proteins, SIR-2.1 and HCF-1, also interacted significantly more with DAF-16 when daf-2 was defective (Supplementary Table 2), but were below the threshold for inclusion in Fig. 1e. These data help benchmark the co-IP and proximity labelling approaches.

In our introduction, we have now explained in detail the advantages and disadvantages offered by proximity labelling compared to other methods, especially co-immunoprecipitation (co-IP). With TurboID proximity labelling once an interactor is biotinylated, the biotin is covalently bound to it. The femtomolar affinity of biotin for streptavidin permits extraction, affinity purification and washing under harsh, denaturing conditions. By contrast, co-IP requires gentle extraction and washing to ensure complexes remain intact. TurboID requires proximity between the bait protein and the interactor: in cytoplasm biotinylation occurs when interactors are within ~10 nm of the bait. Unlike with co-IP, proximity labelling does not necessarily mean two proteins are together in a complex. This is both an advantage and a disadvantage, since *in vivo* proximity has biological meaning. TurboID identifies interactors *in vivo* – the biotinylation occurs in intact cells and animals. Co-IP, by contrast, is performed from *ex vivo* extracts, and is therefore more prone to artefacts. TurboID is able to detect transient or weak interactions – the estimate is that an interaction that lasts 10 minutes is sufficient for biotinylation by TurboID. TurboID highlights proximity at any point in the life of the bait protein, from biogenesis to degradation, which can be both useful and a distraction. In summary, TurboID and co-IP each have strengths and weaknesses, and offer complementary approaches.

2. Among the potential interactors of DAF-16 found by proximity labeling, which ones are downstream effectors of insulin signaling, for instance, transcriptional targets of DAF-16? Are they up- or down-regulated in the *daf-2* mutant?

To address this question, we performed RT-qPCR analysis of selected interactors using total RNA levels extracted from wild type (N2), *daf-2* and *daf-2; daf-16* mutants. We

present these data in a supplementary figure (Supplementary Fig 1d). For most interactors we observed either small or no changes in mRNA levels. For XPC-1, a DNA damage recognition and repair factor, we observed an ~2-fold increase in mRNA levels in *daf-2* mutants that is DAF-16-dependent. For ZFH-2, we observed a small decrease (down to 0.75 of WT) in mRNA levels in *daf-2* mutants that is only partly DAF-16 dependent. For PAR-1, we observed approximately a halving of mRNA levels in *daf-2* mutants that is DAF-16-dependent. Thus, not only does PAR-1 inhibit DAF-16, but DAF-16 can also inhibit *par-1* expression.

3. The investigation of PAR-1 is more in-depth than that of ZFH-2, which in turn is more in-depth than that of MTCL-1, a presumed microtubule-crosslinking factor. I suggest that the MTCL-1 section be removed. Does the degradation of MTCL-1 lead to impairment of microtubules? Is this verified? Additionally, there is no verification data showing physical interaction between MTCL-1 and DAF-16, or that the regulation of DAF-16 by MTCL-1 is direct. DAF-16 is extremely sensitive to stress of all kinds; it translocates readily to the nucleus when a worm is removed from its most comfortable conditions. Nuclear accumulation of DAF-16 induced by MTCL-1 AID could be interpreted as DAF-16 responding to stress caused by impairment of the microtubule network.

As suggested by our reviewer, we have removed the MTCL-1 data. We replaced it by additional data about how PAR-1 functions to regulate DAF-16 (see below).

4. An *in vitro* kinase assay is needed to test whether PAR-1 can directly phosphorylate DAF-16 at S249.

As suggested by our reviewer, we have performed *in vitro* kinase assays using purified proteins to show that PAR-1 and one of its human orthologs, MARK2, can directly phosphorylate DAF-16 at S249. These data are shown in Fig. 6. Briefly, PAR-1 and MARK2 can phosphorylate synthesized peptides whose sequence is centred around DAF-16 S249. Moreover, a peptide designed from the corresponding region of FOXO3a, a human ortholog of DAF-16, is also phosphorylated by MARK2 and PAR-1. These results suggest that the regulation we report in *C. elegans* is conserved in humans. As a positive control we used purified MEX-5, which is a known substrate of PAR-1. As negative controls we used peptides in which we changed the serine residues to alanine. As expected, this abolished signal in our *in vitro* kinase assay.

Minor Issues:

1. "Interactors" identified by proximity labeling should be "potential interactors."

We have made this change as requested.

2. The authors show that ZFH-2 negatively regulates *mtl-1*. Does ZFH-2 regulate *mtl-1* expression directly or through DAF-16? How can we reconcile the lifespan phenotype caused by ZFH-2 degradation?

We used Western blots and qPCR to probe this question. While knockdown of ZFH-2::AID caused a tenfold increase in MTL-1::mNG::FLAG (Fig. 2f), it did not significantly change *mtl-1* mRNA levels as determined by qPCR, even when *daf-16* was defective (Supplementary Fig. 2f). This contrasts with the 500-fold increase in *mtl-1* mRNA levels that we observed when *daf-2* is defective, an increase that is DAF-16-dependent, as reported previously (Supplementary Fig. 1e). These data suggest that the increase in MTL-1 protein levels in ZFH-2 knockdown animals is not DAF-16-dependent.

To address the question of how ZFH-2::AID knockdown regulates lifespan we performed qPCR on two other DAF-16 targets, *sod-3* (encoding a super oxide dismutase), and *adh-1* (encoding an alcohol dehydrogenase). *sod-3* expression more than halved when ZFH-2 was knocked down, and this decrease was partly dependent on DAF-16. *adh-1* expression was not altered when ZFH-2 was knocked down in *daf-16(+)* animals, but was reduced slightly in a *daf-16* deletion mutant. Reduced SOD-3 expression is consistent with reduced stress resistance, but understanding why *zfh-2* knockdown animals live shorter would require further study, for example by examining the batteries of genes whose expression is regulated by this DNA binding protein.

3. *zfh-2*, *par-1*, and *mtcl-1* all encode multiple isoforms. Schematic diagrams showing the knock-in sites of mSc::AID::3*FLAG tag would be more informative than textual descriptions.

We have added Supplementary figures in which we show where we knocked tags into each of these genes. For *daf-16* (alias R13H8.1), see Supplementary Fig 1a; for *zfh-2* (alias ZC123.3) see Supplementary Fig 2e; for *par-1* (alias H39E23.1), see Supplementary Fig 3a.

4. The gene name of *mtcl-1* in Wormbase is *tag-241*.

We renamed *tag-241* as *mtcl-1* with permission from Tim Schedl, the curator for gene names for the *C. elegans* community. The name has not yet been made official. We have however removed the *mtcl-1* data.

Reviewer #2 (Remarks to the Author):

Artan et al present an interesting application of the turboID proximity labeling system, which they use to identify direct protein interactors of DAF-16. *daf-16* is an evolutionary conserved transcription factor of great general interest, and a protein that has been intensively studied in the context of *C. elegans* aging for over thirty years. Previous systematic studies of *daf-16* interactors have focused primarily on its direct and indirect transcriptional targets (e.g Murphy et al 2003 10.1038/nature01789), and a small number of *daf-16* regulators identified using genetic techniques. A systematic method for identifying direct DAF-16 interactors was previously developed, involving IP pull-down of interactors using DAF-16 as bait, followed by mass spectrometry (Riedel 2013 10.1038/ncb2720). turboID has several potential advantages over this previous IP

pulldown method, including increased sensitivity and the ability to better capture in vivo interactions that may not persist ex vivo. These potential advantages provide good motivation for the application of turboID to study DAF-16, as done here by Artan et al.

We thank our reviewer for their supportive comments.

The authors succeed in identifying several novel DAF-16 interactors. In particular, they quite extensively validate PAR-1 as a physiologic regulator of DAF-16, going so far as to identify the specific position on DAF-16 that is modified during PAR-1's post-translational regulation of DAF-16. The authors also identify several other proteins that appear to regulate DAF-16 localization. The authors confirm the physiologic relevance of these interactors by demonstrating that their knockdown modulates DAF-16's influence on lifespan.

The authors present their findings clearly using sound arguments. Where the current manuscript falls bit short is in clarifying exactly how the field benefits from the discovery of these additional mechanistic interactors of DAF-16. Knockdown of DAF-16 has, for twenty years, been known to differentially regulate double-digit percentages of all genes expression (Murphy et al 2003), most of which remain poorly understood. The post-translational regulation of DAF-16 is already known to be rather complex, involving acetylation (Ao-Lin Hsu et al PLOS Genetics 2012 10.1371/journal.pgen.1002948), ubiquitination (Heimbucher et al Cell Metabolism 2015 10.1016/j.cmet.2015.06.002) and a variety of other interactors (Yen, Antioxid Redox Sig 2011, 10.1089/ars.2010.3490). How does the addition of four or five additional regulatory mechanisms to the existing set further our understanding of DAF-16 and its physiologic role? How are the newly discovered protein interactors unique, different, or important in the context of the large body of existing knowledge of DAF-16 function? Clarification of these questions would also help demonstrate that turboID is a technology that can provide novel biological insights.

Major points

1. The mechanistic characterization of DAF-16 interactions appears solid (with a few small caveats below). The lifespan experiments establish the physiologic relevance of the physical interactions described, which are mediated by the well-known influence of DAF-16 localization on organismal gene-expression and aging. However, the authors are mostly silent on what the new regulatory factors teach us beyond the simple fact of their existence. The manuscript would be more compelling to a broader audience were it clearer what generalizable knowledge the paper contributes.

Our reviewer raises thought-provoking questions. We have sought to address them by adding data and by re-writing the discussion to highlight better why we think our findings are significant.

To generalize our findings, we examined if human MARK kinases could phosphorylate human FOXOs. Humans express four MARK kinases and four FOXO transcription factors (TFs). All four human FOXOs conserve the amino acid sequence motif surrounding the S249 phosphorylation site in DAF-16. We have added *in vitro* biochemistry data that suggest that not only does PAR-1 directly phosphorylate DAF-16, but mammalian MARK kinases can also directly phosphorylate FOXO transcription factors. Briefly, we show that purified recombinant human MARK2 and *C. elegans* PAR-1 can phosphorylate *in vitro* peptides from FOXO3 and DAF-16 that contain the sequence motif surrounding S249 at the S249 residue. These data suggest the regulation we describe in worms will likely apply in humans.

We also add more context for our findings in the discussion. PAR-1/MARK kinases have not previously been linked to insulin signalling or to DAF-16/FOXO regulation. They are, however, key regulators of cell polarity, microtubule dynamics, asymmetric cell division, neuronal polarity and epithelial organization. In mammals, the four MARK kinases are expressed very broadly, and mouse mutants defective in different MARK kinases have a range of phenotypes, including reduced adiposity, defective gluconeogenesis, and altered fertility, learning and memory, growth and metabolism. Their functional importance is likely even broader, since different MARK kinases can, to a certain extent at least, substitute for each other.

The regulation of MARK kinases is under intense scrutiny. It is known for example, that LKB1 (Liver kinase B1), TAO-1 (thousand and one) kinases, and DAPK (death-associated protein kinase) activate MARK kinases, whereas GSK3 β and aPKC inhibit them. Thus, our findings also suggest routes by which kinases that regulate MARK kinases can indirectly regulate FOXO activity. The picture that emerges suggests that PAR-1/MARK kinases could provide a brake to keep DAF-16/FOXO activity in check in these kinase networks. For example, LKB1 activates not only MARK kinases, which would inhibit FOXOs, but also AMPK, which activate FOXOs.

Both MARK kinases and FOXO transcription factors are also implicated in devastating human diseases, notably neurodegeneration (e.g. Alzheimer's) and cancer, and are drug targets. This makes further study of the newly discovered link between these pathways reported in this paper potentially relevant in these disease contexts.

2. The authors find that DAF-16 protein abundance is upregulated 2-fold by PAR-1 knockdown (Fig. 3 h-i). The authors should clarify whether they believe this is a regulatory mechanism separate from PAR-1's activity to promote phosphorylation at DAF-16 residue S249, or not. They state that "not all effects of PAR-1 knockdown are mediated via phosphorylation of S249" but then show data that the change in protein abundance seems to require the wild-type s249 residue. If the influence of PAR-1 on DAF-16 abundance requires an intact phosphorylation site, then the change in DAF-16 abundance affected by PAR-1 would then seem mediated via phosphorylation, no? What do the authors think is going on here?

To explicitly address this question, we used our gene edited strains to compare levels of DAF-16, DAF-16 S249A, and DAF-16 S249D with or without PAR-1 knockdown. The levels of all three versions of DAF-16 doubled when we knocked down PAR-1 (Supplementary Fig. 5e, f). Thus, PAR-1 regulation of DAF-16 levels does not depend on phosphorylation of residue S249. To investigate if PAR-1 regulation of DAF-16 is evident at the mRNA level, we performed RT-qPCR of *daf-16* with and without PAR-1 knockdown. We did not observe any change in *daf-16* mRNA levels when we knocked down PAR-1 (Supplementary Fig. 3d), suggesting PAR-1 may regulate DAF-16 protein levels post-translationally. In summary, PAR-1 inhibits DAF-16 in multiple ways. These data may help explain why knockdown of PAR-1 more strongly extends lifespan than the edited *daf-16 S249A* allele.

3. In some places, the author's logic in providing a mechanistic rationale for identifying specific proteins as DAF-16 interactions seem a bit weak. For example, in the case of APT-9 the authors state that "These proteins are implicated in vesicular traffic and microtubule function, hinting at their possible involvement in keeping DAF-16 outside the nucleus when insulin signaling is high." However, it is not clear to this reviewer how APT-9's role in vesicular trafficking would suggest that it should keep DAF-16 outside the nucleus. Do the authors believe that all proteins involved in vesicular trafficking should be involved in DAF-16 regulation?

We agree that without further experiments it is only possible to speculate whether proximity labelling of APT-9 by DAF-16 is biologically relevant. Some recent work does point to DAF-16 being recruited to endosomes. What is striking is that we did not detect enrichment of a host of proteins involved in vesicular traffic, but rather a select few: APT-9, TMYB-1 (target of Myb1) and SEC-15. Similarly, we identified a group of proteins involved in axo-dendritic traffic in neurons, the kinesin UNC-104, SYD-1 and SNPN-1/SNAPIN, which is part of the BLOC-1 (biogenesis of lysosome-related organelle-1) complex. In the discussion we have sought to point out potential entry points for further study while being brief.

4. The logic for selecting specific hits from the turboID analysis for follow-up experimental is currently unclear. In retrospect, the choice of PAR-1 and other interactors appears well-justified, but the authors could not have known this would be the case in advance. What criteria were used to prioritize PAR-1 and the other interactors studied, over other hits with similar statistical support in the turboID data? A clearer rationale would provide important context for the Par-1 results, while also helping future users of turboID will be in a similar position trying to select promising targets from a list of turboID interactors.

For example: in Fig. 1d there are many statistically significant hits with large effect sizes that are unlabeled and remain unmentioned in the manuscript. Looking at the supplementary tables, many potentially interesting interactors are present, for example genes involved in transcription (eg F57B10.8, taf-3) or nuclear import (eg MEL28). Is

there a reason these targets were not followed up upon? For example, maybe these hits would appear for any transcription factor characterized via turboID, and so are not interesting DAF-16-specific interactors?. Other hits from the turboID experiment that remain unmentioned include SYD-1 (protein Q86NH1), which seems to be an interesting gene involved in neuronal development and G5ECF6, a transposon-derived protein. Are these DAF-16 interactors worth following up on?

Our reviewer raises several good points. The choices we made were pragmatic, and in our revised manuscript we have more clearly explained them.

We selected PAR-1 for two main reasons. First, since PAR-1 is a kinase, its interaction with DAF-16 made testable predictions, i.e. altered phosphorylation. Second, and perhaps more important, we thought that linking insulin signalling with PAR-1/MARK kinase signalling would bring together two important areas of research. MARK kinases are expressed very broadly in mammals, and mouse mutants defective in different MARK kinases have a range of phenotypes, including reduced adiposity, defective gluconeogenesis, and altered fertility, learning and memory, growth and metabolism. MARK kinases are also implicated in devastating human diseases, notably neurodegeneration (e.g. Alzheimer's) and cancer, and are drug targets. Their regulation is under intense scrutiny, so that providing evidence that MARK kinases regulate FOXO, which themselves are associated with multiple human diseases (including cancer and neurodegeneration) seemed worthwhile.

We chose the ZFH-2 protein because it is conserved and broadly expressed and, together with its human orthologs, has been linked to obesity. Lesions in human ZFH-2 orthologs have also been repeatedly associated with cancer. Importantly for us, the family is understudied.

Many other potential DAF-16 interactors we highlight may, however be worth following up. It is worth mentioning that we have performed TurboID proximity labelling with several other transcription factors. While some of the interactors highlighted by DAF-16 are also highlighted by these other transcription factors, others are specific to DAF-16. The DNA replication / repair factors XPC-1, SEL-13/ZNF830 and RFC-4 are in this group; DAF-16 may bind to these factors and regulate their activity – a hypothesis worth investigating.

5. On a related note, the authors highlight nuclear proteins whose interactions increase in *daf-2* mutants as well as cytoplasmic proteins whose interactions decrease in *daf-2* mutants. Such proteins fit the narrative of DAF-2 activating DAF-16 primarily by promoting its nuclear localization. However, is it the case that not a single protein showed an opposite-than-expected effect—e.g, a cytoplasmic protein that increased in biotinylation after DAF-2 knockdown, or a nuclear protein that decreased in biotinylation after DAF-2 knockdown? Such proteins might be very interesting.

Generally, the proteins that show an opposite-than-expected effect are contaminants that are present also in our no TurboID controls, at least with the strict cutoff values we used – namely $\log_2 \geq 3$ and $p \leq 0.05$ for nuclear DAF-16, and $\log_2 \leq -2$ and $p \leq 0.05$ for cytoplasmic DAF-16. One exception is SAMS-5, an S-adenosylmethionine synthetase that generates S-adenosylmethionine (SAM), a methyl donor for transmethylation. Interestingly, knockdown of a paralog of SAMS-5, SAMS-1, extends lifespan (Chen et al 2024), although no DAF-16 link has been examined. Whether DAF-16 helps bring SAMS-5 to the nucleus, to locally elevate SAM levels to promote methylation e.g. of histones, requires further analyses. Our TurboID experiment does identify the histone lysine methyltransferases SET-26/MLL5 and ZFP-1/MLLT10 as enriched close to DAF-16.

6. One more area that seems to be relatively unexplored in the manuscript is the potential for DAF-16 to have functional interactions with cytoplasmic proteins that are not regulators of DAF-16. The authors highlight several cases where cytoplasmic proteins modify DAF-16 and serve to regulate DAF-16, but is this the exclusive role of DAF-16 interactions in the cytoplasm? Could the causal relationship be reversed for any of the new identified cytoplasmic biotinylation targets—could they be regulated by DAF-16, and not vice-versa?

The reviewer is absolutely right. DAF-16 could be regulating some interactors, rather than the other way around. This holds both for interactors in cytoplasm (in wild animals) and the nucleus (in *daf-2* mutants). We have added a paragraph to highlight this possibility in the discussion. There are already hints of this in the literature. For example, FOXO3a directly binds the ataxia-telangiectasia mutated kinase ATM in tissue culture cells to regulate its DNA repair activity. Perhaps DAF-16 also regulates the activity of several other DNA repair factors e.g. XPC-1, SEL-13/ZNF830 and RFC-4, with which it interacts. Probing further such potential roles would be interesting and worthwhile.

Reviewer #3 (Remarks to the Author):

Artan and coauthors reported the use of proximity labeling of DAF-16/FOXO in *C. elegans*. This study is interesting and provides new insights into the function of DAF-16/FOXO and the use of cytoplasmic and nuclear conditions to identify DAF-16 interactors enriched in different subcellular locations and their functions. The reviewer was asked by the editor to evaluate the quality of the proteomic experiments, but many details are missing in the Methods section, making it difficult for the reviewer to properly assess the quality of the experiments.

We thank our reviewer for their supportive comments. We apologize for inadvertently omitting some details about how we did experiments. We have sought to address these omissions, as we explain below. We have added these to relevant sections of the manuscript, usually to the methods section.

1. The authors cited their previously published manuscripts (refs 15 and 16) as references for the proximity labeling method; however, the reviewer still finds certain aspects unclear. For example, why was the WT control used instead of the TurboID control described in the references?

To date we have performed proximity labelling with about 20 *C. elegans* proteins, but we continue to debate how best to control these experiments. For this paper, the critical test used to select interactors for further study involved comparing the biotinylation signal from animals expressing DAF-16::TurboID in *daf-2* mutants and wild type animals. We included controls that express no TurboID to ensure that we effectively recognized and removed the 'crapome', that is proteins that non-specifically bind the affinity column, under the precise extraction conditions we were using.

The main difficulty with using the TurboID controls described in the references our reviewer refers to is matching the expression levels of control and experimental TurboID fusions. Ideally, the negative control for the *daf-2* mutant experiments would be a biologically inactive nuclear protein, e.g. a GFP-NLS-TurboID, that is expressed in the same cells at exactly the same levels as DAF-16-TurboID, since DAF-16::TurboID is nuclear in *daf-2* mutants. For the *daf-2(+)* experiments, an ideal negative control would be a cytoplasmically-localized GFP-TurboID, again expressed in the same cells and at similar levels as DAF-16::TurboID. Matching both the expression pattern and expression levels of experimental and control TurboID fusions is, however, surprisingly difficult. When we have made artificial operons, which allow us to match expression pattern, we have found that the intrinsic stability of control and experimental fusions are invariably very different: the control fusions are typically expressed at significantly higher levels, which can then lead to false negatives.

Like other omics approaches, proximity labelling provides a useful way to generate hypotheses, but these then need to be tested experimentally. For both PAR-1 and ZFH-2 that led us to perform co-IP experiments, to show that these proteins are in a complex with DAF-16.

What developmental stage of the worms was used for each strain, and why? How many plates or worms were used? What was the total amount of protein used for enrichment? What imputation method was used?

We collected synchronized, well-fed L4/young adults grown at 21 °C from uncrowded plates to extract proteins for all our proximity labelling experiments. The main reason for using staged animals was to minimize, as far as possible, variation across biological replicates. We used L4s/young adults as these are post-mitotic in the soma, and have few if any developing embryos in their uteri. They are also large, and therefore yield more protein. L4/young adults are the most commonly used stage for 'omic' studies in *C. elegans*, unless there are specific reasons to use a different stage. Future work could,

however, extend his work by studying how the DAF-16 interactome changes as animals age, for example by investigating synchronized 9-day old adults.

We processed ~ 0.5 g of animals for each sample, which we obtained by adding 20,000 eggs to each of 15–20 9 cm NGM plates. To normalize across experiments, we adjusted the protein concentration of extracts so that we always used 12.5 mg of total protein as input for the enrichment.

Missing LFQ values were imputed in Log space by drawing random values from a normal distribution (estimated from sample LFQ intensity distributions and shifted by -1.8 standard deviations with a width of 0.3 standard deviations) after filtering out contaminants and proteins with less than 2 razor and unique peptides. For pairwise comparisons the Log₂ fold change and mean LFQ were calculated and plotted against each other (using an MAplot).

2. Similarly, additional details are needed on how DAF-16 phosphorylation was determined. What was the amount of input lysate? What was the amount of purified DAF-16? How much was loaded onto the gel? Please also provide the MS2 spectra for the selected phosphorylation sites.

We used 10 mg of input lysate to affinity purify DAF-16-HA for analysing DAF-16 phosphorylation. We did not measure the amount of DAF-16 we purified, as the protein we obtained by boiling the anti-HA beads in sample buffer was not pure. We loaded the entire eluate from the beads onto the polyacrylamide gel, and cut bands at the appropriate molecular weight for the major DAF-16 bands. The mass spectrometry facility injected 45% of the peptides obtained from every gel slice for mass spectrometry. As requested by our reviewer, we have included the MS2 spectra for all phosphosites identified in Supplementary Table 4.

We have modified the manuscript to include the clarifications requested by our reviewer.

Reviewer #4 (Remarks to the Author):

Thank you for co-reviewing the manuscript!